# Stabilized Proximal-Point Methods for Federated Optimization

**Xiaowen Jiang**
Saarland University and CISPA*
xiaowen.jiang@cispa.de

**Anton Rodomanov**
CISPA*
anton.rodomanov@cispa.de

**Sebastian U. Stich**
CISPA*
stich@cispa.de

## Abstract

In developing efficient optimization algorithms, it is crucial to account for communication constraints—a significant challenge in modern Federated Learning. The best-known communication complexity among non-accelerated algorithms is achieved by DANE, a distributed proximal-point algorithm that solves local subproblems at each iteration and that can exploit second-order similarity among individual functions. However, to achieve such communication efficiency, the algorithm requires solving local subproblems sufficiently accurately resulting in slightly sub-optimal local complexity. Inspired by the hybrid-projection proximal-point method, in this work, we propose a novel distributed algorithm S-DANE. Compared to DANE, this method uses an auxiliary sequence of prox-centers while maintaining the same deterministic communication complexity. Moreover, the accuracy condition for solving the subproblem is milder, leading to enhanced local computation efficiency. Furthermore, S-DANE supports partial client participation and arbitrary stochastic local solvers, making it attractive in practice. We further accelerate S-DANE and show that the resulting algorithm achieves the best-known communication complexity among all existing methods for distributed convex optimization while still enjoying good local computation efficiency as S-DANE. Finally, we propose adaptive variants of both methods using line search, obtaining the first provably efficient adaptive algorithms that could exploit local second-order similarity without the prior knowledge of any parameters.

## 1 Introduction

Federated learning is a rapidly emerging large-scale machine learning framework that allows training from decentralized data sources (e.g. mobile phones or hospitals) while preserving basic privacy and security [43, 24, 31]. Developing efficient federated optimization algorithms becomes one of the central focuses due to its direct impact on the effectiveness of global machine learning models.

One of the key challenges in modern federated optimization is to tackle communication bottlenecks [32]. The large-scale model parameters, coupled with relatively limited network capacity and unstable client participation, often make communication highly expensive. Therefore, the primary efficiency metric of a federated optimization algorithm is the total number of communication rounds required to reach a desired accuracy level. If two algorithms share equivalent communication complexity, their local computation efficiency becomes the second important metric.

The seminal algorithm DANE [57] is an outstanding distributed optimization method. It achieves the best-known deterministic communication complexity among existing non-accelerated algorithms (on the server side) [22]. This efficiency primarily hinges upon a mild precondition regarding the Second-order Dissimilarity $\delta$. In numerous scenarios, like statistical learning for generalized model [19] and

---

*CISPA Helmholtz Center for Information Security, Saarbrücken, Germany

semi-supervised learning [7], $\delta$ tends to be relatively small. However, to ensure such fast convergence, DANE requires each iteration subproblem to be solved with sufficiently high accuracy. This leads to sub-optimal local computation effort across the communication rounds, which is inefficient in practice. FEDRED [22] improves this weakness by using double regularization. However, this technique is only effective when using gradient descent as the local solver but cannot easily be combined with other optimization methods. For instance, applying local accelerated gradient or second-order methods cannot improve its local computation efficiency. Moreover, it is also unclear how to extend this method to the partial client participation setting relevant to federated learning.

On the other hand, the communication complexities achieved by the current accelerated methods typically cannot be directly compared with those attained by DANE, as they either depend on sub-optimal constants or additional quantities such as the number of clients $n$. The most relevant and state-of-the-art algorithm ACC-EXTRAGRADIENT [33] achieves a better complexity in terms of the accuracy $\varepsilon$ but with dependency on the maximum Second-order Dissimilarity $\delta_{\max}$ which can in principle be much larger than $\delta$ (see Remark 2). Unlike most federated learning algorithms, such as FEDAVG [43], this method requires communication with all the devices at each round to compute the full gradient and then assigns one device for local computation. In contrast, FEDAVG and similar algorithms perform local computations on parallel and utilize the standard averaging to compute the global model. The follow-up work AccSVRS [40] applies variance reduction to ACC-EXTRAGRADIENT which results in less frequent full gradient updates. However, the communication complexity incurs a dependency on $n$ which is prohibitive for cross-device setting [24] where the number of clients can be potentially very large. Thus, there exists no accelerated federated algorithm that is uniformly better than DANE in terms of communication complexity.

**Contributions.** In this work, we aim to develop federated optimization algorithms that can achieve the best communication complexity while retaining efficient local computation. To this end, we first revisit the simple proximal-point method on a single machine. The accuracy requirement for solving the subproblem defined in this algorithm is slightly sub-optimal. Drawing inspiration from hybrid projection-proximal point method for finding zeroes of a maximal monotone operator [59], we observe that using a more stabilized prox-center improves the accuracy requirement. We make the following contributions:

- We develop a novel federated optimization algorithm S-DANE that achieves the best-known communication complexity (for non-accelerated methods) while also enjoying improved local computation efficiency over DANE [57].

- We develop an accelerated version of S-DANE based on the Monteiro-Svaiter acceleration [46]. The resulting algorithm ACC-S-DANE achieves the best-known communication complexity among all existing methods for distributed convex optimization.

- Both algorithms support partial client participation and arbitrary stochastic local solvers, making them attractive in practice for federated optimization.

- We provide a simple analysis for both algorithms. We derive convergence estimates that are continuous in the strong convexity parameter $\mu$.

- We propose adaptive variants of both algorithms using line-search in the full client participation setting. The resulting methods achieve the same communication complexity (up to a logrithmic factor) as non-adaptive ones without requiring knowledge of the similarity constant.

- We illustrate strong practical performance of our proposed methods in experiments.

See also Table 1 for a summary of the main complexity results in the full-participation setting.

**Related Work.** Moreau first proposed the notion of the proximal approximation of a function [47]. Based on this operation, Martinet developed the first proximal-point method [42]. This method was first accelerated by Güller [17], drawing the inspiration from Nesterov's Fast gradient method [49]. Later, Lin et al. [41] introduced the celebrated CATALYST framework that builds upon Güller's acceleration. Using CATALYST acceleration, a large class of optimization algorithms can directly achieve faster convergence. In a similar spirit, Doikov and Nesterov [12] propose contracting proximal methods that can accelerate higher-order tensor methods. While Güller's acceleration has been successfully applied to many settings, its local computation is sub-optimal. Specifically, when minimizing smooth convex functions, a logarithmic dependence on the final accuracy is

| Algorithm | # Vectors comm per round | General convex | | μ-strongly convex | | Guarantee |
|---|---|---|---|---|---|---|
| | | # comm rounds | # local gradient queries | # comm rounds | # local gradient queries | |
| Scaffnew [44] [a] | $n$ | $\frac{LD^2}{\varepsilon}$ [b] | $1$ [b] | $\sqrt{\frac{L}{\mu}}\log\frac{D^2+H_0^2/(\mu L)}{\varepsilon}$ | $\sqrt{\frac{L}{\mu}}$ | in expectation |
| SONATA [60] [c] | $n$ | unknown | unknown | $\frac{\delta_{\max}}{\mu}\log\frac{D^2}{\varepsilon}$ | $-$ [d] | deterministic |
| DANE [57] | $n$ | $\frac{\delta D^2}{\varepsilon}$ | $\sqrt{\frac{L}{\delta}}\log\frac{LD^2}{\varepsilon}$ | $\frac{\delta}{\mu}\log\frac{D^2}{\varepsilon}$ | $\sqrt{\frac{L}{\delta}}\log\left(\frac{L}{\mu}\log\frac{D^2}{\varepsilon}\right)$ | deterministic |
| FedRed [22] | $n$ | $\frac{\delta D^2}{\varepsilon}$ | $\frac{L}{\delta}$ | $\frac{\delta}{\mu}\log\frac{D^2}{\varepsilon}$ | $\frac{L}{\delta}$ | in expectation |
| S-DANE (**this work**, Alg. 1) | $n$ | $\frac{\delta D^2}{\varepsilon}$ | $\sqrt{\frac{L}{\delta}}$ | $\frac{\delta}{\mu}\log\frac{D^2}{\varepsilon}$ | $\sqrt{\frac{L}{\delta}}$ | deterministic |
| Inexact Acc-SONATA [61] [c] | $n$ | unknown | unknown | $\sqrt{\frac{\delta_{\max}}{\mu}}\log\frac{\delta_{\max}}{\mu}\log\frac{D^2}{\varepsilon}$ | $\sqrt{\frac{L}{\mu}}\log\frac{D^2}{\varepsilon}$ | deterministic |
| Acc-Extragradient [33] [c] | $n$ | $\sqrt{\frac{\delta_{\max}D^2}{\varepsilon}}$ | $\sqrt{\frac{L}{\delta_{\max}}}$ | $\sqrt{\frac{\delta_{\max}}{\mu}}\log\frac{D^2}{\varepsilon}$ | $\sqrt{\frac{L}{\delta_{\max}}}$ | deterministic |
| Catalyzed SVRP [28] [e] | $\begin{cases}1 & \text{w.p. } 1-\frac{1}{n},\\ n & \text{w.p. }\frac{1}{n}\end{cases}$ | unknown | unknown | $\left(n+n^{\frac{3}{4}}\sqrt{\frac{L}{\mu}}\right)\log\frac{L}{\mu}\log\frac{D^2}{\varepsilon}$ | $-$ [f] | in expectation |
| AccSVRS [40] [e c] | | | | $\left(n+n^{\frac{3}{4}}\sqrt{\frac{\delta}{\mu}}\right)\log\frac{D^2}{\varepsilon}$ | $\frac{1}{n^{1/4}}\sqrt{\frac{L}{\delta}}$ | in expectation |
| Acc-S-DANE (**this work**, Alg. 2) | $n$ | $\sqrt{\frac{\delta D^2}{\varepsilon}}$ | $\sqrt{\frac{L}{\delta}}$ | $\sqrt{\frac{\delta}{\mu}}\log\frac{D^2}{\varepsilon}$ | $\sqrt{\frac{L}{\delta}}$ | deterministic |

[a] For SCAFFNEW and FEDRED, the column '# comm rounds' represents the expected number of total communications required to reach $\varepsilon$ accuracy. The column '# local gradient queries' is replaced with the expected number of local steps between two communications.

[b] The general convex result of SCAFFNEW is established in Theorem 11 in the RANDPROX paper [8]. We assume that $\mathbf{h}_{i,0}=\nabla f_i(\mathbf{x}^0)$ and estimate $H_0^2:=\frac{1}{n}\sum_{i=1}^n\|\mathbf{h}_{i,0}-\nabla f_i(\mathbf{x}^\star)\|\le L^2D^2$. Then the best $p$ is of order 1.

[c] SONATA, INEXACT ACC-SONATA, ACC-EXTRAGRADIENT and ACCSVRS only need to assume strong convexity of $f$.

[d] Exact proximal local steps are used in SONATA

[e] CATALYZED SVRP and ACCSVRS aim at minimizing a different measure which is the total amount of information transmitted between the server and the clients. Their iteration complexity is equivalent to the communication rounds in our notations. We refer to Remark 7 for details.

[f] Khaled and Jin [28] assume exact evaluations of the proximal operator for the convenience of analysis.

Table 1: Summary of the worst-case convergence behaviors of the considered distributed optimization methods (in the BigO-notation) assuming each $f_i$ is $L$-smooth and $\mu$-convex with $\mu\le\Theta(\delta)$, where $\delta$, $\delta_{\max}$, $\zeta^2$ are defined in (2), Remark 2 and (3), and $D:=\|\mathbf{x}^0-\mathbf{x}^\star\|$. The '*# local gradient queries*' column represents the number of gradient oracle queries required between two communication rounds to achieve the corresponding complexity, assuming **the most efficient local first-order algorithms are used**. The column *'Guarantee'* means whether the convergence guarantee holds in expectation or deterministically. The suboptimality $\varepsilon$ is defined via $\|\hat{\mathbf{x}}^R-\mathbf{x}^\star\|^2$ and $f(\hat{\mathbf{x}}^R)-f^\star$ for strongly-convex and general convex functions where $\hat{\mathbf{x}}^R$ is a certain output produced by the algorithm after $R$ number of communications.

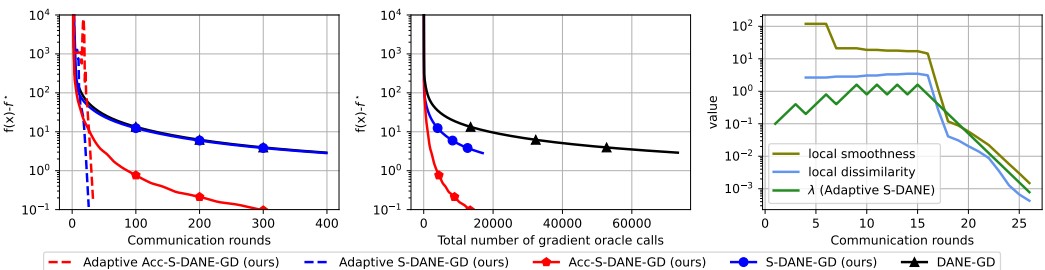

Figure 1: Comparison of S-DANE and ACC-S-DANE with DANE for solving a convex quadratic minimization problem. All three methods use GD as the local solver. S-DANE has improved local computation efficiency than DANE while ACC-S-DANE further improves the communication complexity. Finally, the adaptive variants can leverage local dissimilarities to achieve better performance. (The definitions of local smoothness and dissimilarity can be found in Section 6.)

incurred in its local computation complexity [12]. Solodov and Svaiter [59] proposed a HYBRID PROJECTION-PROXIMAL POINT method that allows significant relaxation of the accuracy condition for the proximal-point subproblems. More recent works such as ADAPTIVE CATALYST [21] and RECAPP [5] successfully get rid of the additional logarithmic factor for accelerated proximal-point methods as well.

Another type of acceleration based on the proximal extra-gradient method was introduced by Monteiro and Svaiter [46]. This method is more general in the sense that it allows arbitrary local solvers and the convergence rates depend on the these solvers. For instance, under convexity and Lipschitz second-order derivative, the rate can be accelerated to $\mathcal{O}(1/k^{3.5})$ by using Newton-type method. Moreover, when the gradient method is used, Monteiro-Svaiter Acceleration recovers the rate of Güller's acceleration and its accuracy requirement for the inexact solution is weaker. For minimizing smooth convex functions, one gradient step is enough for approximately solving the local subproblem [48]. This technique has been applied to centralized composite optimization, known as gradient sliding [35, 36, 33]. A comprehensive study of acceleration can be found in [14].

We defer the literature review on distributed and federated optimization algorithms to Appendix A.

## 2 Problem Setup and Background

We consider the following distributed minimization problem:

$$\min_{\mathbf{x} \in \mathbb{R}^d} \Big\{ f(\mathbf{x}) := \frac{1}{n} \sum_{i=1}^{n} f_i(\mathbf{x}) \Big\}, \tag{1}$$

where each function $f_i \colon \mathbb{R}^d \to \mathbb{R}$ is $\mu$-strongly convex[2] for some $\mu \geq 0$. We focus on the standard federated setting where the functions $\{f_i\}$ are distributed among $n$ devices. The server coordinates the global optimization procedure among the devices. In each communication round, the server broadcasts certain information to the clients. The clients, in turn, perform local computation in parallel based on their own data and transmit the resulting local models back to the server to update the global model.

**Main objective:** Given the high cost of establishing connections between the server and the clients, our paramount objective is to minimize the number of required communication rounds to achieve the desired accuracy level. This represents a central metric in federated contexts, as outlined in references such as [43, 27]. **Secondary objective:** Efficiency in local computation represents another pivotal metric for optimization algorithms. For instance, if two algorithms share equivalent communication complexity, the algorithm with less local computational complexity is the more favorable choice.

**Notation:** We abbreviate $[n] := \{1, 2, \ldots, n\}$. For a set $A$ and an integer $1 \leq s \leq |A|$, we use $\binom{A}{s}$ to denote the power set comprised of all $s$-element subsets of $A$. Everywhere in this paper, $\|\cdot\|$ denotes the standard Euclidean norm (or the corresponding spectral norm for matrices). We assume problem (1) has a solution which we denote by $\mathbf{x}^\star$; the corresponding optimal value is denoted by $f^\star$. For a set $S \in \binom{[n]}{s}$, we use $f_S := \frac{1}{s} \sum_{i \in S} f_i$ to denote the average function over this set. We use $r$ to denote the index of the communication round and $k$ to denote the index of the local step. Finally, we use the superscript and subscript to denote the global and local models, respectively; for instance, $\mathbf{x}^r$ represents the global model at round $r$ while $\mathbf{x}_{i,r}$ is the local model computed by device $i$ at round $r$.

### 2.1 Proximal-Point Methods on Single Machine

In this section, we provide a brief background on proximal-point methods [47, 42, 54, 51], which are the foundation for many distributed optimization algorithms.

**Proximal-Point Method.** Given an iterate $\mathbf{x}_k$, the method defines $\mathbf{x}_{k+1}$ to be an (approximate) minimizer of the proximal-point subproblem:

$$\mathbf{x}_{k+1} \approx \arg\min_{\mathbf{x} \in \mathbb{R}^d} \Big\{ F_k(\mathbf{x}) := f(\mathbf{x}) + \frac{\lambda}{2} \|\mathbf{x} - \mathbf{x}_k\|^2 \Big\}, \tag{2}$$

for an appropriately chosen parameter $\lambda \geq 0$. This parameter allows for a trade-off between the complexity of each iteration and the rate of convergence. If $\lambda = 0$, the subproblem in each iteration is as difficult as solving the original problem because no regularization is applied. However, as $\lambda$ increases, more regularization is added, simplifying the subproblem. For example, for a convex function $f$, the proximal-point method guarantees $f(\bar{\mathbf{x}}_K) - f^\star \leq \mathcal{O}\big(\frac{\lambda}{K} \|\mathbf{x}_0 - \mathbf{x}^\star\|^2\big)$, where $\bar{\mathbf{x}}_K := \frac{1}{K} \sum_{k=1}^{K} \mathbf{x}_k$ [51, 5]. However, to achieve such a convergence rate, the subproblem (2) has to be solved to a fairly high accuracy [54, 5]. For instance, the accuracy condition should either depend on the target accuracy $\varepsilon$, or increase with $k$: $\|\nabla F_k(\mathbf{x}_{k+1})\| = \mathcal{O}(\frac{\lambda}{k} \|\mathbf{x}_{k+1} - \mathbf{x}_k\|)$ [58]. Indeed, when $f$ is Lipschitz-smooth and the standard gradient descent is used as a local solver, the number of gradient steps required to solve the subproblem has a logarithmic dependence on the iteration counter $k$. The same issue also arises when considering accelerated proximal-point methods [12, 17].

**Stabilized Proximal-Point Method.** One of the key insights that we use in this work is the observation that using a different prox-center makes the accuracy condition of the subproblem weaker.

---

[2] If $\mu = 0$, then $f_i$ is assumed to be simply convex.

---

**Algorithm 1** S-DANE: Stabilized DANE

---

1: **Input:** $\lambda > 0$, $\mu \geq 0$, $s \in [n]$, $\mathbf{x}^0 = \mathbf{v}^0 \in \mathbb{R}^d$.
2: **for** $r = 0, 1, 2 \ldots$ **do**
3:      Sample $S_r \in \binom{[n]}{s}$ uniformly at random without replacement.
4:      **for each device** $i \in S_r$ **in parallel do**
5:          $\mathbf{x}_{i,r+1} \approx \arg\min_{\mathbf{x} \in \mathbb{R}^d} \left\{ F_{i,r}(\mathbf{x}) := f_i(\mathbf{x}) + \langle \nabla f_{S_r}(\mathbf{v}^r) - \nabla f_i(\mathbf{v}^r), \mathbf{x} \rangle + \frac{\lambda}{2} \|\mathbf{x} - \mathbf{v}^r\|^2 \right\}$.
6:      $\mathbf{x}^{r+1} = \frac{1}{s} \sum_{i \in S_r} \mathbf{x}_{i,r+1}$.
7:      $\mathbf{v}^{r+1} := \arg\min_{\mathbf{x} \in \mathbb{R}^d} \left\{ \frac{1}{s} \sum_{i \in S_r} [\langle \nabla f_i(\mathbf{x}_{i,r+1}), \mathbf{x} \rangle + \frac{\mu}{2} \|\mathbf{x} - \mathbf{x}_{i,r+1}\|^2] + \frac{\lambda}{2} \|\mathbf{x} - \mathbf{v}^r\|^2 \right\}$.

---

The *stabilized proximal-point method* defines

$$
\begin{aligned}
\mathbf{x}_{k+1} &\approx \arg\min_{\mathbf{x}} \left\{ F_k(\mathbf{x}) := f(\mathbf{x}) + \frac{\lambda}{2} \|\mathbf{x} - \mathbf{v}_k\|^2 \right\}, \\
\mathbf{v}_{k+1} &= \arg\min_{\mathbf{x}} \left\{ \langle \nabla f(\mathbf{x}_{k+1}), \mathbf{x} \rangle + \frac{\mu}{2} \|\mathbf{x} - \mathbf{x}_{k+1}\|^2 + \frac{\lambda}{2} \|\mathbf{x} - \mathbf{v}_k\|^2 \right\},
\end{aligned}
\tag{3}
$$

where $\lambda \geq 0$ is a parameter of the method and $\mu \geq 0$ is the strong-convexity constant of $f$. This algorithm updates the prox-center $\mathbf{v}_k$ by performing an additional gradient step in each iteration. For instance, when $\mu = 0$, the prox-center is updated as $\mathbf{v}_{k+1} = \mathbf{v}_k - \frac{1}{\lambda} \nabla f(\mathbf{x}_{k+1})$, which is often referred to as an *extra-gradient update*. The stabilized proximal-point method has the same convergence rate as the original method (2) but requires only that $\|\nabla F_k(\mathbf{x}_{k+1})\| \leq \mathcal{O}(\lambda \|\mathbf{x}_{k+1} - \mathbf{v}_k\|)$. As a result, there is no extra logarithmic factor of $k$ in the oracle complexity estimate when $f$ is $L$-smooth. Specifically, by setting $\lambda = \Theta(L)$, the previous condition can be satisfied by choosing $\mathbf{x}_{k+1}$ as the result of one gradient step from $\mathbf{v}_k$ [48]. This shows that the stabilized proximal-point method has a better overall oracle complexity than the standard proximal-point method (c.f. Theorem 1 for the special case $n = 1$). It is worth noting that the former algorithm originates from the hybrid projection-proximal point algorithm [59] designed for solving the more general problem of finding zeroes of a monotone operator. In this work, we apply this algorithm in the distributed setting ($n \geq 2$).

## 2.2 Distributed Proximal-Point Methods

The proximal-point method can be adapted to solve the distributed optimization problem (1). This is the idea behind FEDPROX [39]. It replaces the global proximal step (2) by $n$ subproblems defined as $\mathbf{x}_{i,r+1} := \arg\min_{\mathbf{x}} \{ f_i(\mathbf{x}) + \frac{\lambda}{2} \|\mathbf{x} - \mathbf{x}^r\|^2 \}$, which can be solved independently on each device, followed by the averaging step $\mathbf{x}^{r+1} = \frac{1}{n} \sum_{i=1}^n \mathbf{x}_{i,r+1}$. Here we switch the notation from $k$ to $r$ to highlight that one iteration of the proximal-point method corresponds to a communication round in this setting. To ensure convergence, FEDPROX has to use a large $\lambda$ that depends on the target accuracy as well as the heterogeneity among $\{f_i\}$, which slows down the communication efficiency [39]. DANE [57] improves this by incorporating a drift correction term into the subproblem:

$$
\mathbf{x}_{i,r+1} := \arg\min_{\mathbf{x}} \left\{ \tilde{F}_{i,r}(\mathbf{x}) := f_i(\mathbf{x}) + \langle \nabla f(\mathbf{x}^r) - \nabla f_i(\mathbf{x}^r), \mathbf{x} \rangle + \frac{\lambda}{2} \|\mathbf{x} - \mathbf{x}^r\|^2 \right\}.
\tag{4}
$$

Consequently, DANE allows to choose a much smaller $\lambda$ in the algorithm. Moreover, it can exploit second-order similarity and achieve the best-known communication complexity among non-accelerated methods [22]. However, as in the original proximal-point method, the subproblem needs to be solved sufficiently accurately leading to an extra logarithmic factor in the oracle complexity estimate. To overcome this problem, we propose new algorithms described in the following section.

## 3 Stabilized DANE

We now describe S-DANE (Alg. 1), our proposed federated proximal-point method that employs stabilized prox-centers in its subproblems. During each communication round $r$, the server samples a subset of clients uniformly at random and sends $\mathbf{v}^r$ to these clients. Then the server collects $\nabla f_i(\mathbf{v}^r)$ from these clients, computes $\nabla f_{S_r}(\mathbf{v}^r)$ and sends $\nabla f_{S_r}(\mathbf{v}^r)$ back to them. Each device in the set then calls an arbitrary local solver (which can be different on each device) to approximately solve its local subproblem. Finally, each device transmits $\nabla f_i(\mathbf{x}_{i,r+1})$ and $\mathbf{x}_{i,r+1}$ back to the server which then aggregates these points and computes the new global model.

As DANE, S-DANE can also achieve communication speed-up if the functions among devices are similar to each other. This is formally captured by the following assumption.

**Definition 1** (Second-order Dissimilarity). *Let $f_1, \ldots, f_n : \mathbb{R}^d \to \mathbb{R}$ be functions, and let $s \in [n]$, $\delta_s \geq 0$. Then, $\{f_i\}_{i=1}^n$ are said to have $\delta_s$-SOD (of size $s$) if for any $\mathbf{x}, \mathbf{y} \in \mathbb{R}^d$ and any $S \in \binom{[n]}{s}$, it holds that*

$$\frac{1}{s} \sum_{i \in S} \|\nabla h_i^S(\mathbf{x}) - \nabla h_i^S(\mathbf{y})\|^2 \leq \delta_s^2 \|\mathbf{x} - \mathbf{y}\|^2, \tag{5}$$

*where $h_i^S := f_S - f_i$ and $f_S := \frac{1}{s} \sum_{i \in S} f_i$.*

Definition 1 quantifies the dissimilarity between any $s$ functions and their average, i.e., the "internal" variation between any $s$ functions. Clearly, $\delta_1 = 0$, and, when $s = n$, we recover the standard notion of second-order dissimilarity introduced in prior works:

**Definition 2** ($\delta$-SOD [28, 40, 22]). *$\delta$-SOD $:= \delta_n$-SOD of size $n$.*

When each function $f_i$ is twice continuously differentiable, a simple sufficient condition for (5) is that $\frac{1}{s} \sum_{i \in S} \|\nabla^2 h_i^S(\mathbf{x})\|^2 \leq \delta_s^2$ for any $\mathbf{x} \in \mathbb{R}^d$. However, this is not a necessary condition (see [22] for more details).

The quantity $V(\mathbf{x}, \mathbf{y})$ in the left-hand side of (5) can be interpreted as the variance of the gradient difference estimator $\nabla f_{\hat{i}}(\mathbf{x}) - \nabla f_{\hat{i}}(\mathbf{y})$, where $\hat{i}$ is chosen uniformly at random from $S$. In particular, it can be rewritten as $V(\mathbf{x}, \mathbf{y}) = \frac{1}{s} \sum_{i \in S} \|\nabla f_i(\mathbf{x}) - \nabla f_i(\mathbf{y})\|^2 - \|\nabla f_S(\mathbf{x}) - \nabla f_S(\mathbf{y})\|^2$. If each function $f_i$ is $L_i$-smooth, then $\delta_s \leq (\frac{1}{s} \sum_{i \in S} L_i^2)^{1/2}$ for any $s \in [n]$. However, in general, condition (5) is weaker than assuming that each $f_i$ is Lipschitz-smooth.

**Full Client Participation.**    We first consider the cross-silo setting where all the clients are highly reliable ($s = n$). This is typically the case with organizations and institutions having strong computing resources and stable network connection [24].

**Theorem 1.** *Consider Algorithm 1 with $s = n$. Let $f_i \colon \mathbb{R}^d \to \mathbb{R}$ be $\mu$-convex with $\mu \geq 0$ for any $i \in [n]$. Assume that $\{f_i\}_{i=1}^n$ have $\delta$-SOD. Let $\lambda = 2\delta$ and suppose that, for any $r \geq 0$, we have*

$$\sum_{i=1}^n \|\nabla F_{i,r}(\mathbf{x}_{i,r+1})\|^2 \leq \frac{\lambda^2}{4} \sum_{i=1}^n \|\mathbf{x}_{i,r+1} - \mathbf{v}^r\|^2. \tag{6}$$

*Then, for any $R \geq 1$, it holds that[3]*

$$f(\bar{\mathbf{x}}^R) - f^\star \leq \frac{\mu D^2}{2[(1 + \frac{\mu}{2\delta})^R - 1]} \leq \frac{\delta D^2}{R},$$

*where $\bar{\mathbf{x}}^R := \frac{1}{\sum_{r=1}^R p^r} \sum_{r=1}^R p^r \mathbf{x}^r$ for $p := 1 + \frac{\mu}{\lambda}$, and $D := \|\mathbf{x}^0 - \mathbf{x}^\star\|$. To obtain $f(\bar{\mathbf{x}}^R) - f^\star \leq \varepsilon$ for a given $\varepsilon > 0$, it thus suffices to perform $R = \mathcal{O}\left(\frac{\delta + \mu}{\mu} \log(1 + \frac{\mu D^2}{\varepsilon})\right)$ communication rounds.*

Theorem 1 provides the convergence guarantee for S-DANE in terms of the number of communication rounds. Note that the rate is continuous in $\mu$.

**Remark 2.** *Some previous works express complexity estimates in terms of another constant, $\delta_{\max}$, defined by the inequality $\|\nabla h_i(\mathbf{x}) - \nabla h_i(\mathbf{y})\| \leq \delta_{\max} \|\mathbf{x} - \mathbf{y}\|$ holding for any $\mathbf{x}, \mathbf{y} \in \mathbb{R}^d$ and any $i \in [n]$, where $h_i = f - f_i$. (See for instance the second line in Table 1). Note that our $\delta$ is always not larger than $\delta_{\max}$, and can in principle be much smaller (up to $\sqrt{n}$ times).*

The proven communication complexity is the same as that of DANE [22]. However, the accuracy condition is milder. Specifically, to achieve the same guarantee, DANE requires $\sum_{i=1}^n \|\nabla \tilde{F}_{i,r}(\mathbf{x}_{i,r+1})\|^2 \leq \mathcal{O}(\frac{\delta^2}{r^2} \sum_{i=1}^n \|\mathbf{x}_{i,r+1} - \mathbf{x}^r\|^2)$, where $\tilde{F}_{i,r}$ is defined as in (4), which incurs an $r^2$ overhead in the denominator, as in the general discussion on proximal-point methods in Section 2.1. The next corollary shows that local computations in S-DANE could be computationally very efficient.

---

[3]Here, for $\mu = 0$, the expression after the first inequality should be understood as the corresponding limit when $\mu \to 0$; $\mu > 0$, which is exactly the expression after the final inequality. The same remark applies to all other similar results.

**Corollary 3.** *Consider the same setting as in Theorem 1. Further, assume that each $f_i$ is L-smooth. To ensure (6) with a certain first-order algorithm, each device $i$ needs to perform at most $\mathcal{O}(\sqrt{\frac{L}{\delta}})$ computations of $\nabla f_i$ at each round $r$.*

**Remark 4.** *Particular examples of algorithms that could be used to achieve the result from Corollary 3 are OGM-OG by Kim and Fessler [30] and the accumulative regularization method by Lan et al. [37], both designed for the fast minimization of the gradient norm. For the standard Gradient Method (GM), the required number of oracle calls is $\mathcal{O}(\frac{L}{\delta})$. The standard Fast Gradient Method (FGM) [49] can further decrease the complexity to $\mathcal{O}(\sqrt{\frac{L}{\mu+\delta}}\log\frac{L}{\delta})$ (see Remark 18 for details). Thus, each device can run a constant number of standard local (F)GM steps to approximately solve their subproblems in S-DANE.*

**Partial Client Participation.** Next, we turn our attention to the cross-device setting where a large number of clients (typically mobile phones) have either unstable network connection or weak computational power [24]. In such scenarios, the server typically cannot expect all the clients to be able to participate in the communication at each round. Furthermore, the clients may typically be asked to communicate only once during the whole training and are stateless [27]. Therefore, we now consider S-DANE with partial client participation and without storing any states on devices.

To prove convergence, it is necessary to assume a certain level of dissimilarity among clients. Here, we use the same assumption as in [27] to measure the gradient variance.

**Definition 3** (Bounded Gradient Variance [27]). *Let $f_1, \ldots, f_n : \mathbb{R}^d \to \mathbb{R}$ be functions, and let $\zeta \geq 0$. We say that $\{f_i\}_{i=1}^n$ have $\zeta$-BGV if, for any $\mathbf{x} \in \mathbb{R}^d$ and $f := \frac{1}{n}\sum_{i=1}^n f_i$, it holds that*

$$\frac{1}{n}\sum_{i=1}^n \|\nabla f_i(\mathbf{x}) - \nabla f(\mathbf{x})\|^2 \leq \zeta^2. \tag{7}$$

Definition 3 is similar to the classical notion of uniformly bounded variance used in the context of classical stochastic gradient methods [3].

We also need the following assumption which complements Definition 1.

**Definition 4** (External Dissimilarity). *Let $f_1, \ldots, f_n : \mathbb{R}^d \to \mathbb{R}$ be functions, and let $s \in [n]$, $\Delta_s \geq 0$. Then, $\{f_i\}_{i=1}^n$ are said to have $\Delta_s$-ED (of size $s$) if, for any $\mathbf{x}, \mathbf{y} \in \mathbb{R}^d$ and any $S \in \binom{[n]}{s}$, we have*

$$\|\nabla m_S(\mathbf{x}) - \nabla m_S(\mathbf{y})\| \leq \Delta_s\|\mathbf{x} - \mathbf{y}\|, \tag{8}$$

*where $m_S := f - f_S$ and $f_S := \frac{1}{s}\sum_{i \in S} f_i$.*

Compared to Definition 1, the new Definition 4 quantifies the "external" variation of any $s$ functions w.r.t. the original function $f$. When each $f_i$ is twice continuously differentiable, (8) is equivalent to $\|\nabla^2 m_S(\mathbf{x})\| \leq \Delta_s$ for any $\mathbf{x} \in \mathbb{R}^d$. If each $f_i$ is L-smooth, then $\Delta_s \leq L$ for any $s \in [n]$. Therefore, using both Assumptions 1 and 4 is still weaker than assuming that each $f_i$ is L-smooth.

In what follows, we work with a new second-order dissimilarity measure defined as the sum $\delta_s + \Delta_s$. Note that $\delta_1 + \Delta_1 = \delta_{\max}$ and $\delta_n + \Delta_n = \delta$.

**Theorem 5.** *Consider Algorithm 1. Let $f_i : \mathbb{R}^d \to \mathbb{R}$ be $\mu$-convex with $\mu \geq 0$ for any $i \in [n]$ and let $n \geq 2$. Assume that $\{f_i\}_{i=1}^n$ have $\delta_s$-SOD, $\Delta_s$-ED and $\zeta$-BGV. Let $\lambda = \frac{4(n-s)}{s(n-1)}\frac{\zeta^2}{\varepsilon} + 2(\delta_s + \Delta_s)$, and suppose that, for any $r \geq 0$, we have*

$$\frac{1}{s}\sum_{i \in S_r}\|\nabla F_{i,r}(\mathbf{x}_{i,r+1})\|^2 \leq \frac{\lambda^2}{4}\frac{1}{s}\sum_{i \in S_r}\|\mathbf{x}_{i,r+1} - \mathbf{v}^r\|^2. \tag{9}$$

*Then, to ensure that $\mathbb{E}[f(\bar{\mathbf{x}}^R)] - f(\mathbf{x}^\star) \leq \varepsilon$ for a given $\varepsilon > 0$, it suffices to perform at most the following number of communication rounds:*

$$R = \Theta\left(\left[\frac{\delta_s + \Delta_s + \mu}{\mu} + \frac{n-s}{n-1}\frac{\zeta^2}{s\varepsilon\mu}\right]\log\left(1 + \frac{\mu D^2}{\varepsilon}\right)\right) \leq \Theta\left(\frac{(\delta_s + \Delta_s)D^2}{\varepsilon} + \frac{n-s}{n-1}\frac{\zeta^2 D^2}{s\varepsilon^2}\right),$$

*where $\bar{\mathbf{x}}^R := \frac{1}{\sum_{r=1}^R p^r}\sum_{r=1}^R p^r\mathbf{x}^r$ with $p := 1 + \frac{\mu}{\lambda}$, and $D := \|\mathbf{x}^0 - \mathbf{x}^\star\|$.*

---

**Algorithm 2** ACC-S-DANE

---

1: **Input:** $\lambda > 0$, $\mu \geq 0$, $\mathbf{x}^0 = \mathbf{v}^0 \in \mathbb{R}^d$, $s \in [n]$.
2: Set $A_0 = 0$, $B_0 = 1$.
3: **for** $r = 0, 1, 2, \ldots$ **do**
4:     Find $a_{r+1} > 0$ from the equation $\lambda = \frac{(A_r + a_{r+1})B_r}{a_{r+1}^2}$.
5:     $A_{r+1} = A_r + a_{r+1}$, $B_{r+1} = B_r + \mu a_{r+1}$.
6:     $\mathbf{y}^r = \frac{A_r}{A_{r+1}}\mathbf{x}^r + \frac{a_{r+1}}{A_{r+1}}\mathbf{v}^r$.
7:     Sample $S_r \in \binom{[n]}{s}$ uniformly at random without replacement.
8:     **for each device** $i \in S_r$ **in parallel do**
9:         $\mathbf{x}_{i,r+1} \approx \arg\min_{\mathbf{x}\in\mathbb{R}^d}\big\{F_{i,r}(\mathbf{x}) := f_i(\mathbf{x}) + \langle \nabla f_{S_r}(\mathbf{y}^r) - \nabla f_i(\mathbf{y}^r), \mathbf{x}\rangle + \frac{\lambda}{2}\|\mathbf{x} - \mathbf{y}^r\|^2\big\}$.
10:    $\mathbf{x}^{r+1} = \frac{1}{s}\sum_{i\in S_r}\mathbf{x}_{i,r+1}$.
11:    $\mathbf{v}^{r+1} = \arg\min_{\mathbf{x}\in\mathbb{R}^d}\big\{\frac{a_{r+1}}{s}\sum_{i\in S_r}[\langle \nabla f_i(\mathbf{x}_{i,r+1}), \mathbf{x}\rangle + \frac{\mu}{2}\|\mathbf{x} - \mathbf{x}_{i,r+1}\|^2] + \frac{B_r}{2}\|\mathbf{x} - \mathbf{v}^r\|^2\big\}$.

---

Theorem 5 provides the communication complexity of S-DANE with client sampling and arbitrary (deterministic) local solvers. The rate is again continuous in $\mu$. Compared with the previous case of $s = n$, the efficiency now depends on the gradient variance $\zeta$. Note that this error term gets reduced when $s$ increases. Specifically, to achieve the $\mathcal{O}(\log\frac{1}{\varepsilon})$ and $\mathcal{O}(\frac{1}{\varepsilon})$ rates, it suffices to ensure that $s = \Theta\big(\frac{n\zeta^2}{\zeta^2 + n\varepsilon(\delta_s + \Delta_s)}\big)$. Notably, the algorithm can reach any target accuracy even when $n \to \infty$.

Observe that the accuracy requirement (9) is the same as (6). Therefore, the discussions therein are valid in the partial-participation setting as well. In particular, if each $f_i$ is $L$-smooth, then the number of oracle calls to $\nabla f_i$ required at each round could be as small as $\mathcal{O}(\sqrt{\frac{L}{\lambda}})$ (see Corollary 3). At the same time, it is also possible to use a stochastic optimization algorithm as a local solver (for more details, see Section C.3.1).

## 4 Accelerated S-DANE

In this section, we present the accelerated version of S-DANE, ACC-S-DANE (Alg. 2), that achieves a better communication complexity compared to the basic method. For simplicity, we only consider the full-participation setting and defer the partial participation to Appendix D.3.

**Theorem 6.** *Consider Algorithm 2 with $s = n$. Let $f_i : \mathbb{R}^d \to \mathbb{R}$ be $\mu$-convex with $\mu \geq 0$ for any $i \in [n]$. Assume that $\{f_i\}_{i=1}^n$ have $\delta$-SOD ($\delta > 0$). Let $\lambda = 2\delta$ and suppose that, for any $r \geq 0$, we have $\sum_{i=1}^n \|\nabla F_{i,r}(\mathbf{x}_{i,r+1})\|^2 \leq \delta^2 \sum_{i=1}^n \|\mathbf{x}_{i,r+1} - \mathbf{y}^r\|^2$. If $\mu \leq 8\delta$, then, for any $R \geq 1$,*

$$f(\mathbf{x}^R) - f^\star \leq \frac{2\mu D^2}{\big[\big(1 + \sqrt{\frac{\mu}{8\delta}}\big)^R - \big(1 - \sqrt{\frac{\mu}{8\delta}}\big)^R\big]^2} \leq \frac{4\delta D^2}{R^2},$$

*where $D := \|\mathbf{x}^0 - \mathbf{x}^\star\|$. Otherwise, $f(\mathbf{x}^R) - f^\star \leq \frac{4\delta D^2}{(1+\sqrt{\frac{\mu}{8\delta}})^{2(R-1)}}$ for any $R \geq 1$. To ensure that $f(\mathbf{x}^R) - f^\star \leq \varepsilon$ for a given $\varepsilon > 0$, it thus suffices to perform $R = \mathcal{O}\big(\sqrt{\frac{\delta+\mu}{\mu}}\log(1 + \sqrt{\frac{\min\{\mu,\delta\}D^2}{\varepsilon}})\big)$ communication rounds.*

Let us consider the most interesting regime when $\mu \leq 8\delta$. Comparing Theorems 1 and 6, we see that ACC-S-DANE essentially extracts the square root of the corresponding communication complexity of S-DANE by improving it from $\tilde{\mathcal{O}}(\frac{\delta}{\mu})$ to $\tilde{\mathcal{O}}(\sqrt{\frac{\delta}{\mu}})$ when $\mu > 0$, and from $\mathcal{O}(\frac{\delta D^2}{\varepsilon})$ to $\mathcal{O}(\sqrt{\frac{\delta D^2}{\varepsilon}})$ when $\mu = 0$, while maintaining the same accuracy condition for solving the subproblem. Compared with ACC-EXTRAGRADIENT, the complexity depends on a better constant $\delta$ instead of $\delta_{\max}$.

Note that we can satisfy the accuracy condition in Theorem 6 in exactly the same way as in Corollary 3. In particular, if each $f_i$ is $L$-smooth, each device $i$ needs at most $\mathcal{O}(\sqrt{\frac{L}{\delta}})$ computations of $\nabla f_i$ at each round $r$ when using a fast algorithm for the gradient norm minimization.

Finally, let us highlight that Algorithm 2 gives a distributed framework for a *generic acceleration scheme*, that applies to a large class of local optimization methods—in the same spirit as in the

famous CATALYST [41] framework that applies to the case where $n = 1$. However, in contrast to CATALYST, this stabilized version removes the logarithmic overhead present in the original method. Specifically, when applying Theorem 6 with $n = 1$ and $\lambda = L$ for a smooth convex function $f$, we recover the same rate as CATALYST. The accuracy condition $\|\nabla F_r(\mathbf{x}^{r+1})\| \leq L\|\mathbf{x}^{r+1} - \mathbf{y}^r\|$, or equivalently $\langle \nabla f(\mathbf{x}^{r+1}), \mathbf{y}^r - \mathbf{x}^{r+1} \rangle \geq \frac{1}{2L}\|\nabla f(\mathbf{x}^{r+1})\|^2$ can be achieved with one gradient step $\mathbf{x}^{r+1} := \mathbf{y}^r - \frac{1}{L}\nabla f(\mathbf{y}^r)$ (see Lemma 5 in [48]).

## 5 Dynamic Estimation of Similarity Constant by Line Search

One drawback of Algorithms 1 and 2 is that they require the knowledge of the similarity constant $\delta$ to choose an appropriate value for $\lambda$. This similarity constant is typically unknown in practice and might be difficult to estimate. One effective solution to this problem is to dynamically adjust the coefficient $\lambda$ inside the algorithms by using the classical technique of *line search*.

The basic idea is as follows. The server first picks an arbitrary sufficiently small constant $\tilde{\lambda}$ as an initial approximation to the unknown "correct" value of $\lambda = 2\delta$. Then, at every round, the server sends the current estimate of $\lambda$ to each client asking them to approximately solve their local subproblem. After receiving the corresponding local solutions, the server checks a certain inequality based on the obtained information. If this inequality is satisfied, the server accepts the resulting aggregated solution and goes to the next round while decreasing $\lambda$ in two times (so as to be more optimistic in the future). Otherwise, it increases $\lambda$ in two times, asks the clients to solve their subproblems with the new value of $\lambda$, and checks the inequality again.

The precise versions of Algorithms 1 and 2 with line search for the full-participation setting are presented in Algorithms 3 and 4. Importantly, our adaptive schemes are not just some heuristics but are probably efficient. Specifically, their complexity estimates (in terms of the total number of communication rounds) are exactly the same as those given by Theorems 1 and 6, respectively, up to an extra *additive* logarithmic term of $\log \frac{2\delta}{\tilde{\lambda}}$ (see Theorems 27 and 28).

Another significant advantage of our adaptive algorithms is their ability to exploit *local* similarity, resulting in much stronger practical performance compared to the methods with fixed $\lambda$. We will demonstrate this in the next section.

## 6 Numerical Experiments

In this section, we illustrate the performance of our methods in numerical experiments. The implementation can be found at https://github.com/mlolab/S-DANE.

**Convex quadratic minimization.** We first illustrate the properties of our algorithms as applied to minimizing a simple quadratic function: $f(\mathbf{x}) := \frac{1}{n}\sum_{i=1}^{n} f_i(\mathbf{x})$ where $f_i(\mathbf{x}) := \frac{1}{m}\sum_{j=1}^{m} \frac{1}{2}\langle \mathbf{A}_{i,j}(\mathbf{x} - \mathbf{b}_{i,j}), \mathbf{x} - \mathbf{b}_{i,j} \rangle$ where $\mathbf{b}_{i,j} \in \mathbb{R}^d$ and $\mathbf{A}_{i,j} \in \mathbb{R}^{d \times d}$. The experimental details can be found in Appendix F.1. From Figure 1, we see that S-DANE converges as fast as DANE in terms of communication rounds, but with much fewer local gradient oracle calls. ACC-S-DANE achieves the best performance among the three methods. We also test S-DANE and DANE with the same fixed number of local steps. The result can be seen in Figure E.1 where S-DANE is again more efficient. Finally, we report the strong performances of two adaptive variants (Algorithms 3 and 4 with initial $\tilde{\lambda} = 10^{-3}$). We see from Figure 1 that the method can automatically change $\lambda$ to adapt to the local second-order dissimilarity. (We use $\frac{\|\nabla f(\mathbf{v}^{r+1}) - \nabla f(\mathbf{v}^r)\|}{\|\mathbf{v}^{r+1} - \mathbf{v}^r\|}$ and $\sqrt{\frac{\frac{1}{n}\sum_{i=1}^{n}\|\nabla h_i(\mathbf{v}^{r+1}) - \nabla h_i(\mathbf{v}^r)\|^2}{\|\mathbf{v}^{r+1} - \mathbf{v}^r\|^2}}$ to approximate the local smoothness and dissimilarity.)

**Strongly-convex polyhedron feasibility problem.** We now consider the problem of finding a feasible point $\mathbf{x}^\star$ inside a polyhedron: $P = \cap_{i=1}^{n} P_i$, where $P_i = \{\mathbf{x} : \langle \mathbf{a}_{i,j}, \mathbf{x} \rangle \leq \mathbf{b}_{i,j}, \forall j = 1, \ldots, m_i\}$ and $\mathbf{a}_{i,j}, \mathbf{b}_{i,j} \in \mathbb{R}^d$. Each individual function is defined as $f_i := \frac{n}{m}\sum_{j=1}^{m_i}[\langle \mathbf{a}_{i,j}, \mathbf{x} \rangle - \mathbf{b}_{i,j}]_+^2$ where $\sum_{i=1}^{n} m_i = m$. We use $m = 10^5$ and $d = 10^3$. We first generate $\mathbf{x}^\star$ randomly from a sphere with radius $10^6$. We then follow [55] to generate $(\mathbf{a}_{i,j}, \mathbf{b}_{i,j})$ such that $\mathbf{x}^\star$ is a feasible point of $P$ and the initial point of all optimizers is outside the polyhedron. We choose the best $\lambda$ from $\{10^i\}_{i=-3}^{3}$. We first consider the full client participation setting and use $n = s = 10$. We compare our proposed methods with GD, DANE with GD [57], SCAFFOLD with control variate of option I [26], SCAFFNEW [44], FEDPROX with GD [39] and ACC-EXTRAGRADIENT [33]. The

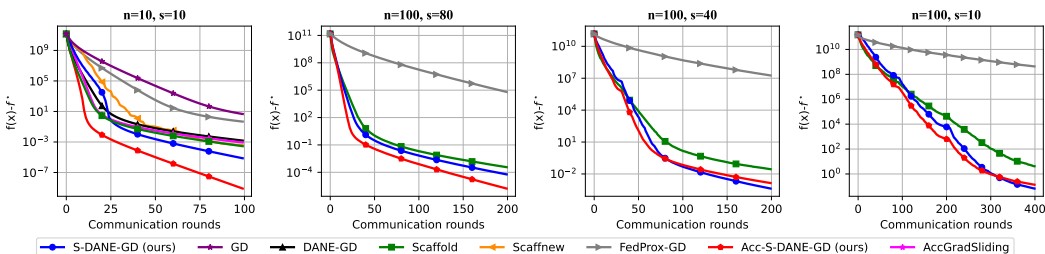

Figure 2: Comparisons of different algorithms for solving the polyhedron feasibility problem.

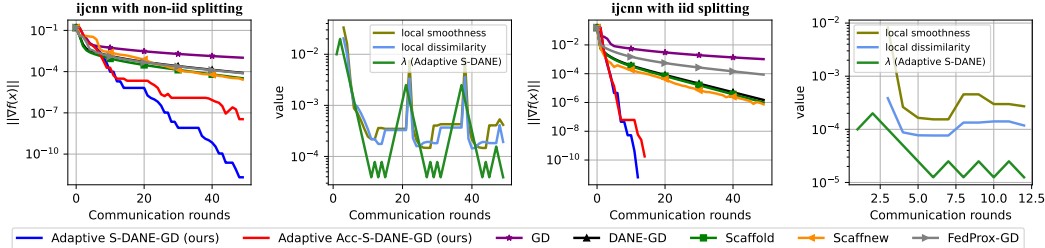

Figure 4: Illustration of the impact of adaptive $\lambda$ used in (ACC-)S-DANE on the convergence of a regularized logistic regression problem on the ijcnn dataset [6].

result is shown in the first plot of Figure 2 where our proposed methods are consistently the best among all these algorithms. We next experiment with client sampling and use $n = 100$. We decrease the number of sampled clients from $s = 80$ to $s = 10$. In addition to our methods, we also report the performances of SCAFFOLD and FEDPROX with client sampling. From the same figure, we see that the improvement of ACC-S-DANE over S-DANE gradually disappears as $s$ decreases.

**Adaptive choice of $\lambda$.** We consider the standard regularized logistic regression: $f(\mathbf{x}) = \frac{1}{n} \sum_{i=1}^{n} f_i(\mathbf{x})$ with $f_i(\mathbf{x}) := \frac{n}{M} \sum_{j=1}^{m_i} \log(1 + \exp(-y_{i,j} \langle \mathbf{a}_{i,j}, \mathbf{x} \rangle)) + \frac{1}{2M} \|\mathbf{x}\|^2$ where $(\mathbf{a}_{i,j}, y_{i,j}) \in \mathbb{R}^{d+1}$ are features and labels and $M := \sum_{i=1}^{n} m_i$ is the total number of data points in the training dataset. We use the ijcnn dataset from LIBSVM [6]. We split the dataset into 10 subsets according to the Dirichlet distribution with $\alpha = 2$ (i.i.d) and $\alpha = 0.2$ (non-i.i.d). From Figure 4, Adaptive (ACC-)S-DANE (Algorithm 3 and 4) converge much faster than the other best-tuned algorithms for both cases. (We set the initial $\tilde{\lambda} = 10^{-4}$ for non-i.i.d and $\lambda = 10^{-5}$ for i.i.d respectively.)

**Deep learning task.** Finally, we consider the multi-class classification tasks with CI-FAR10 [34] using ResNet-18 [18]. The details can be found in Appendix F.2. From Figure 3, we see that S-DANE (DL) 5 reaches 90% accuracy within 50 communication rounds while all the other methods are still below 90% after 80 epochs. The effectiveness of S-DANE on the training of other deep learning models such as Transformer requires further exploration.

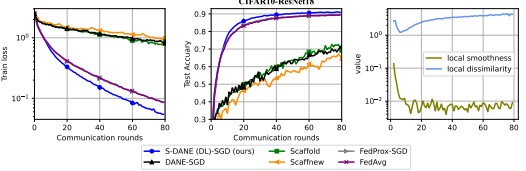

Figure 3: Comparison of S-DANE without control variate against other popular optimizers on multi-class classification tasks with CIFAR10 datasets using ResNet18.

## 7 Conclusion

We have proposed new federated optimization methods (both basic and accelerated) that simultaneously achieve the best-known communication and local computation complexities. The new methods allow partial participation and arbitrary stochastic local solvers, making them attractive in practice. We further equip both algorithms with line search and the resulting schemes can adapt to the local dissimilarity without knowing the corresponding similarity constant. However, we assume that each function $f_i$ is $\mu$-strongly convex in all the theorems. This is stronger than assuming only $\mu$-strongly convexity of $f$, which is used in some prior works. Possible directions for future research include consideration of weaker assumptions as well as empirical and theoretical analyses for non-convex problems.

## Acknowledgments

The authors are grateful to Adrien Taylor and Thomas Pethick for the reference to [59]. The authors are thankful to the anonymous reviewers for their valuable comments and suggestions.

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

# Appendix

## Contents

# A   More Related Work

In the first several years of the development for federated learning algorithms, the convergence guarantees are focused on the smoothness parameter $L$. The de facto standard algorithm for federated learning is FEDAVG. It reduces the communication frequency by doing multiple SGD steps on available clients before communication, which works well in practice [43]. However, in theory, if the heterogeneity among clients is large, then it suffers from the so-called client drift phenomenon [26] and might be worse than centralized mini-batch SGD [63, 29]. Numerous efforts have been made to mitigate this drift impact. FEDPROX adds an additional regularizer to the subproblem of each client based on the idea of centralized proximal point method to limit the drift of each client. However, the communication complexity still depends on the heterogeneity. The celebrated algorithm SCAFFOLD applies drift correction (similar to variance-reduction) to the update of FEDAVG and it successfully removes the impact of the heterogeneity. Afterwards, the idea of drift correction is employed in many other works [15, 62, 38, 1]. SCAFFNEW uses a special choice of control variate[44] and first illustrates the usefulness of taking standard local gradient steps under strongy-convexity, followed with more advanced methods with refined analysis and features such as client sampling and compression [20, 9, 8]. 5gCS [16] uses an approximate proximal-point step at each iteration and derives the convergence rate that is as good as SCAFFNEW, and it also supports client sampling. Later, Sadiev et al. [56] proposed APDA WITH INEXACT PROX that retains the same communication complexity as SCAFFNEW, but further provably reduces the local computation complexity. FedAC [64] applies nesterov's acceleration in the local steps and shows provably better convergence than FEDAVG under certain assumptions.

More recent works try to develop algorithms with guarantees that rely on a potentially smaller constant than $L$. SCAFFOLD first illustrates the usefulness of taking local steps for quadratics under Bounded Hessian Dissimilarity $\delta_{\max}$ [26]. SONATA [60] and its accelerated version [61] prove explicit communication reduction in terms of $\delta_{\max}$ under strong convexity. MIME [27] and CE-LGD [52] work on non-convex settings and show the communication improvement on $\delta_{\max}$ and the latter achieves the min-max optimal rates. ACCELERATED EXTRAGRADIENT SLIDING [33] applies gradient sliding [33] technique and shows communication reduction in terms of $\delta_{\max}$ for strongly-convex and convex functions, the local computation of which is also efficient without logarithmic dependence on the target accuracy, DANE with inexact local solvers [57, 22, 53] has been shown recently to achieve the communication dependency on $\delta$ under convexity and $\delta_{\max}$ under non-convexity. For solving convex problems, the local computation efficiency depends on the target accuracy $\varepsilon$. Otherwise, the accuracy condition for the subproblem should increase across the communication rounds. Hendrikx et al. [19] proposed SPAG, an accelerated method, and prove a better uniform concentration bound of the conditioning number when solving strongly-convex problems. SVRP and CATALYZED SVRP [28] transfer the idea of using the centralized proximal point method to the distributed setting and they achieve communication complexity (with a different notion) w.r.t $\delta$. Lin et al. [40] further improves these two methods either with a better rate or with weaker assumptions based on the ACCELERATED EXTRAGRADIENT SLIDING method. Beznosikov et al. [2] uses compression to reduce the bits required to communicate and the more general problem of Variational Inequalities is considered. Under the same settings but for non-convex optimization, SABER [45] achieves communication complexity reduction with better dependency on $\delta_{\max}$ and $n$. Karagulyan et al. [25] proposed SPAM that allows partial client participation.

**Remark 7.** *Khaled and Jin [28] and Lin et al. [40] consider the total amount of information transmitted between the server and clients as the main metric, which is similar to reducing the total stochastic oracle calls in centralized learning settings. This is a particularly meaningful setting if the server prefers to or has to receive/transmit vectors one by one and can set up communications very fast. The term 'client sampling' in these works refers to sampling one client to do the local computation. However, all the clients still need to participate in the communication from time to time to provide the full gradient information. This is orthogonal to the setup of this work since we assume each device can do the calculation in parallel. In the scenarios where the number of devices is too large such that receiving all the updates becomes problematic, we consider instead the standard partial participation setting.*

# B   Technical Preliminaries

## B.1   Basic Definitions

We use the following definitions throughout the paper.

**Definition 5.** *A differentiable function $f : \mathbb{R}^d \to \mathbb{R}$ is called $\mu$-convex for some $\mu \geq 0$ if for all $\mathbf{x}, \mathbf{y} \in \mathbb{R}^d$,*

$$f(\mathbf{y}) \geq f(\mathbf{x}) + \langle \nabla f(\mathbf{x}), \mathbf{y} - \mathbf{x} \rangle + \frac{\mu}{2} \|\mathbf{x} - \mathbf{y}\|^2. \tag{B.1}$$

*If $f$ is $\mu$-convex, then, for any $\mathbf{x}, \mathbf{y} \in \mathbb{R}^d$, we have (Nesterov [49], Theorem 2.1.10):*

$$\mu \|\mathbf{x} - \mathbf{y}\| \leq \|\nabla f(\mathbf{x}) - \nabla f(\mathbf{y})\|. \tag{B.2}$$

**Definition 6.** *A differentiable function $f : \mathbb{R}^d \to \mathbb{R}$ is called L-smooth for some $L \geq 0$ if for all $\mathbf{x}, \mathbf{y} \in \mathbb{R}^d$,*

$$\|\nabla f(\mathbf{x}) - \nabla f(\mathbf{y})\| \leq L\|\mathbf{x} - \mathbf{y}\|. \tag{B.3}$$

*If $f$ is L-smooth, then, for any $\mathbf{x}, \mathbf{y} \in \mathbb{R}^d$, we have (Nesterov [49], Lemma 1.2.3)*

$$f(\mathbf{y}) \leq f(\mathbf{x}) + \langle \nabla f(\mathbf{x}), \mathbf{y} - \mathbf{x} \rangle + \frac{L}{2}\|\mathbf{y} - \mathbf{x}\|^2. \tag{B.4}$$

**Lemma 8** (Nesterov [49], Theorem 2.1.5)**.** *Let $f : \mathbb{R}^d \to \mathbb{R}$ be convex and L-smooth. Then, for any $\mathbf{x}, \mathbf{y} \in \mathbb{R}^d$, we have*

$$\frac{1}{L}\|\nabla f(\mathbf{x}) - \nabla f(\mathbf{y})\|^2 \leq \langle \nabla f(\mathbf{x}) - \nabla f(\mathbf{y}), \mathbf{x} - \mathbf{y} \rangle. \tag{B.5}$$

## B.2 Useful Lemmas

We frequently use the following helpful lemmas for the proofs.

**Lemma 9.** *For any $\mathbf{x}, \mathbf{y} \in \mathbb{R}^d$ and any $\gamma > 0$, we have*

$$|\langle \mathbf{x}, \mathbf{y} \rangle| \leq \frac{\gamma}{2}\|\mathbf{x}\|^2 + \frac{1}{2\gamma}\|\mathbf{y}\|^2, \tag{B.6}$$

$$\|\mathbf{x} + \mathbf{y}\|^2 \leq (1 + \gamma)\|\mathbf{x}\|^2 + \left(1 + \frac{1}{\gamma}\right)\|\mathbf{y}\|^2. \tag{B.7}$$

**Lemma 10** (Jiang et al. [22], Lemma 14)**.** *Let $\{x_i\}_{i=1}^n$ be a set of vectors in $\mathbb{R}^d$ and let $\bar{\mathbf{x}} := \frac{1}{n}\sum_{i=1}^n \mathbf{x}_i$. Let $\mathbf{v} \in \mathbb{R}^d$ be an arbitrary vector. Then,*

$$\frac{1}{n}\sum_{i=1}^n \|\mathbf{x}_i - \mathbf{v}\|^2 = \|\bar{\mathbf{x}} - \mathbf{v}\|^2 + \frac{1}{n}\sum_{i=1}^n \|\mathbf{x}_i - \bar{\mathbf{x}}\|^2. \tag{B.8}$$

**Lemma 11.** *Let $(F_k)_{k=1}^\infty$ and $(D_k)_{k=0}^\infty$ be two non-negative sequences such that, for any $k \geq 0$, it holds that*

$$F_{k+1} + D_{k+1} \leq qD_k + \varepsilon,$$

*where $q \in (0, 1]$ and $\varepsilon \geq 0$ are some constants. Then for all $K \geq 1$ and $S_K := \sum_{k=1}^K \frac{1}{q^k}$, we have*

$$\frac{1}{S_K}\sum_{k=1}^K \frac{F_k}{q^k} + \frac{1-q}{1-q^K}D_K \leq \frac{1-q}{\frac{1}{q^K}-1}D_0 + \varepsilon.$$

*Proof.* Indeed, for any $k \geq 0$, we have

$$\frac{F_{k+1}}{q^{k+1}} + \frac{D_{k+1}}{q^{k+1}} \leq \frac{D_k}{q^k} + \frac{\varepsilon}{q^{k+1}}.$$

Summing up from $k = 0$ to $k = K - 1$, we get

$$\sum_{k=1}^K \frac{F_k}{q^k} + \frac{D_K}{q^K} \leq D_0 + S_K\varepsilon.$$

Dividing both sides by $S_K$ and substituting $S_K = \frac{1}{1-q}(\frac{1}{q^K} - 1)$, we get the claim. $\square$

**Lemma 12** (c.f. Lemma 2.2.4 in [49])**.** *Let $(A_r)_{r=0}^\infty$ be a non-negative non-decreasing sequence such that $A_0 = 0$ and, for any $r \geq 0$,*

$$A_{r+1} \leq \frac{c(A_{r+1} - A_r)^2}{1 + \mu A_r},$$

*where $c > 0$ and $\mu \geq 0$ are some constants. If $\mu \leq 4c$, then for any $R \geq 0$, we have*

$$A_R \geq \frac{1}{4\mu}\left[\left(1 + \sqrt{\frac{\mu}{4c}}\right)^R - \left(1 - \sqrt{\frac{\mu}{4c}}\right)^R\right]^2 \geq \frac{R^2}{4c}. \tag{B.9}$$

*Otherwise, for any $R \geq 1$, it holds that*

$$A_R \geq \frac{1}{4c}\left(1 + \sqrt{\frac{\mu}{4c}}\right)^{2(R-1)}. \tag{B.10}$$

*Proof.* Denote $C_r = \sqrt{\mu A_r}$. For any $r \geq 0$, it holds that

$$\mu C_{r+1}^2(1 + C_r^2) \leq c(C_{r+1}^2 - C_r^2)^2 \leq c\big(2(C_{r+1} - C_r)C_{r+1}\big)^2 = 4c(C_{r+1} - C_r)^2 C_{r+1}^2.$$

Therefore, for any $r \geq 0$:

$$C_{r+1} - C_r \geq \sqrt{\frac{\mu}{4c}}\sqrt{1 + C_r^2}.$$

When $\mu \leq 4c$, by induction, one can show that, for any $R \geq 0$ (see the proof of Theorem 1 in [50] for details):

$$C_R \geq \frac{1}{2}\left[\left(1 + \sqrt{\frac{\mu}{4c}}\right)^R - \left(1 - \sqrt{\frac{\mu}{4c}}\right)^R\right] \geq \sqrt{\frac{\mu}{4c}}R.$$

When $\mu > 4c$, we have $C_{r+1} - C_r \geq \sqrt{\frac{\mu}{4c}}C_r$. It follows that, for any $R \geq 1$,

$$C_R \geq \left(1 + \sqrt{\frac{\mu}{4c}}\right)^{R-1}C_1 \geq \left(1 + \sqrt{\frac{\mu}{4c}}\right)^{R-1}\sqrt{\frac{\mu}{4c}}.$$

Plugging in the definition of $C_R$, we get the claims. $\qquad\square$

**Lemma 13.** *Let* $\{\mathbf{x}_i\}_{i=1}^n$ *be vectors in* $\mathbb{R}^d$ *with* $n \geq 2$. *Let* $s \in [n]$ *and let* $S \in \binom{[n]}{s}$ *be sampled uniformly at random without replacement. Let* $\bar{\mathbf{x}} := \frac{1}{n}\sum_{i=1}^n \mathbf{x}_i$, $\zeta^2 := \frac{1}{n}\sum_{i=1}^n \|\mathbf{x}_i - \bar{\mathbf{x}}\|^2$, *and* $\bar{\mathbf{x}}_S := \frac{1}{s}\sum_{j \in S}\mathbf{x}_j$. *Then,*

$$\mathbb{E}[\bar{\mathbf{x}}_S] = \bar{\mathbf{x}} \qquad \text{and} \qquad \mathbb{E}[\|\bar{\mathbf{x}}_S - \bar{\mathbf{x}}\|^2] = \frac{n-s}{n-1}\frac{\zeta^2}{s}. \tag{B.11}$$

*Proof.* Let $\binom{n}{m} = \frac{n!}{m!(n-m)!}$ be the binomial coefficient for any $n \geq m \geq 1$. By the definition of $\bar{\mathbf{x}}_S$, we have

$$\bar{\mathbf{x}}_S = \frac{1}{s}\sum_{j \in S}\mathbf{x}_j = \frac{1}{s}\sum_{i=1}^n \mathbb{1}[i \in S]\mathbf{x}_i,$$

where $\mathbb{1}[E]$ denotes the $\{0,1\}$-indicator of the event $E$. Taking the expectation on both sides, we get

$$\mathbb{E}[\bar{\mathbf{x}}_S] = \frac{1}{s}\sum_{i=1}^n \Pr[i \in S]\mathbf{x}_i = \frac{1}{s}\sum_{i=1}^n \frac{\binom{n-1}{s-1}}{\binom{n}{s}}\mathbf{x}_i = \frac{1}{s}\sum_{i=1}^n \frac{s}{n}\mathbf{x}_i = \bar{\mathbf{x}}.$$

Further,

$$\begin{aligned}
\mathbb{E}[\|\bar{\mathbf{x}}_S - \bar{\mathbf{x}}\|^2] &= \mathbb{E}\Big[\frac{1}{s^2}\sum_{i \in S}\sum_{j \in S}\langle \mathbf{x}_i - \bar{\mathbf{x}}, \mathbf{x}_j - \bar{\mathbf{x}}\rangle\Big] \\
&= \mathbb{E}\Big[\frac{1}{s^2}\sum_{i \in S}\|\mathbf{x}_i - \bar{\mathbf{x}}\|^2 + \frac{1}{s^2}\sum_{i,j \in S, i \neq j}\langle \mathbf{x}_i - \bar{\mathbf{x}}, \mathbf{x}_j - \bar{\mathbf{x}}\rangle\Big] \\
&= \mathbb{E}\Big[\frac{1}{s^2}\sum_{i=1}^n \mathbb{1}[i \in S]\|\mathbf{x}_i - \bar{\mathbf{x}}\|^2 + \frac{1}{s^2}\sum_{i,j \in [n], i \neq j}\mathbb{1}[i,j \in S]\langle \mathbf{x}_i - \bar{\mathbf{x}}, \mathbf{x}_j - \bar{\mathbf{x}}\rangle\Big] \\
&= \frac{1}{s^2}\sum_{i=1}^n \Pr[i \in S]\|\mathbf{x}_i - \bar{\mathbf{x}}\|^2 + \frac{1}{s^2}\sum_{i,j \in [n], i \neq j}\Pr[i,j \in S]\langle \mathbf{x}_i - \bar{\mathbf{x}}, \mathbf{x}_j - \bar{\mathbf{x}}\rangle \\
&= \frac{1}{s^2}\sum_{i=1}^n \frac{\binom{n-1}{s-1}}{\binom{n}{s}}\|\mathbf{x}_i - \bar{\mathbf{x}}\|^2 + \frac{1}{s^2}\sum_{i,j \in [n], i \neq j}\frac{\binom{n-2}{s-2}}{\binom{n}{s}}\langle \mathbf{x}_i - \bar{\mathbf{x}}, \mathbf{x}_j - \bar{\mathbf{x}}\rangle \\
&= \frac{\zeta^2}{s} + \frac{s-1}{sn(n-1)}\sum_{i,j \in [n], i \neq j}\langle \mathbf{x}_i - \bar{\mathbf{x}}, \mathbf{x}_j - \bar{\mathbf{x}}\rangle.
\end{aligned}$$

Note that

$$\sum_{i,j \in [n], i \neq j}\langle \mathbf{x}_i - \bar{\mathbf{x}}, \mathbf{x}_j - \bar{\mathbf{x}}\rangle = \sum_{i,j \in [n]}\langle \mathbf{x}_i - \bar{\mathbf{x}}, \mathbf{x}_j - \bar{\mathbf{x}}\rangle - \sum_{i=1}^n \|\mathbf{x}_i - \bar{\mathbf{x}}\|^2 = -n\zeta^2.$$

Thus,

$$\mathbb{E}[\|\bar{\mathbf{x}}_S - \bar{\mathbf{x}}\|^2] = \frac{\zeta^2}{s} - \frac{(s-1)\zeta^2}{s(n-1)} = \frac{n-s}{n-1}\frac{\zeta^2}{s}. \qquad\square$$

**Lemma 14.** *Suppose $\{f_i\}_{i=1}^n$ satisfy $\Delta_s$-ED of size $s \in [n]$ and $\zeta$-BGV with $n \geq 2$. Let $f := \frac{1}{n}\sum_{i=1}^n f_i$ and $f_S := \frac{1}{s}\sum_{i\in S} f_i$, where $S \in \binom{[n]}{s}$ is sampled uniformly at random without replacement. Further, let $\mathbf{y} \in \mathbb{R}^d$ be a fixed point, and let $\mathbf{x}_S \in \mathbb{R}^d$ be a random point defined by a deterministic function of $S$. Then, for any $\gamma > 0$, it holds that*

$$\mathbb{E}_S[f(\mathbf{x}_S) - f_S(\mathbf{x}_S)] \leq \frac{n-s}{n-1}\frac{\gamma\zeta^2}{2s} + \left(\frac{1}{2\gamma} + \frac{\Delta_s}{2}\right)\mathbb{E}_S[\|\mathbf{x}_S - \mathbf{y}\|^2]. \tag{B.12}$$

*Proof.* Let $h_S := f - f_S$. Since $\{f_i\}$ satisfy $\Delta_s$-ED (Definition 4), we have, in view of inequality (B.4),

$$h_S(\mathbf{x}_S) \leq h_S(\mathbf{y}) + \langle \nabla h_S(\mathbf{y}), \mathbf{x}_S - \mathbf{y}\rangle + \frac{\Delta_s}{2}\|\mathbf{x}_S - \mathbf{y}\|^2$$

$$\overset{(B.6)}{\leq} h_S(\mathbf{y}) + \frac{\gamma}{2}\|\nabla h_S(\mathbf{y})\|^2 + \frac{1}{2\gamma}\|\mathbf{x}_S - \mathbf{y}\|^2 + \frac{\Delta_s}{2}\|\mathbf{x}_S - \mathbf{y}\|^2$$

Rearranging and taking the expectation on both sides, we get, for any $\gamma > 0$,

$$\mathbb{E}_S[h_S(\mathbf{x}_S) - h_S(\mathbf{y})] \leq \frac{\gamma}{2}\mathbb{E}_S[\|\nabla h_S(\mathbf{y})\|^2] + \frac{1}{2\gamma}\mathbb{E}_S[\|\mathbf{x}_S - \mathbf{y}\|^2] + \frac{\Delta_s}{2}\mathbb{E}_S[\|\mathbf{x}_S - \mathbf{y}\|^2]$$

$$\overset{(7)}{\leq} \frac{n-s}{n-1}\frac{\gamma\zeta^2}{2s} + \left(\frac{1}{2\gamma} + \frac{\Delta_s}{2}\right)\mathbb{E}_S[\|\mathbf{x}_S - \mathbf{y}\|^2],$$

where the last inequality is due to (B.11). Using the fact that $\mathbb{E}_S[f(\mathbf{y}) - f_S(\mathbf{y})] = 0$, we get the claim. $\square$

**Lemma 15.** *Suppose $\{f_i\}_{i=1}^n$ satisfy $\delta_s$-SOD of size $s \in [n]$. Let $f_S := \frac{1}{s}\sum_{i\in S} f_i$ where $s \in [n]$ and $S \in \binom{[n]}{s}$. Let $\mathbf{v} \in \mathbb{R}^d$ be a fixed point, $\lambda > \delta_s$, and let*

$$F_i(\mathbf{x}) := f_i(\mathbf{x}) + \langle \nabla h_i^S(\mathbf{v}), \mathbf{x}\rangle + \frac{\lambda}{2}\|\mathbf{x} - \mathbf{v}\|^2,$$

*where $h_i^S := f_S - f_i$. Let $\{\mathbf{x}_i\}_{i\in S}$ be a set of points in $\mathbb{R}^d$ (such that $\mathbf{x}_i \approx \arg\min_{\mathbf{x}} F_i(\mathbf{x})$ in the sense that $\|\nabla F_i(\mathbf{x}_i)\|$ is sufficiently small), and let $\bar{\mathbf{x}}_S = \frac{1}{s}\sum_{i\in S}\mathbf{x}_i$. Then,*

$$\frac{1}{s}\sum_{i\in S}\langle \nabla f_i(\mathbf{x}_i) + \nabla h_i^S(\bar{\mathbf{x}}_S), \mathbf{v} - \mathbf{x}_i\rangle - \frac{1}{2\lambda}\left\|\frac{1}{s}\sum_{i\in S}\nabla f_i(\mathbf{x}_i)\right\|^2$$

$$\geq \frac{\lambda - \delta_s}{2}\frac{1}{s}\sum_{i\in S}\|\mathbf{v} - \mathbf{x}_i\|^2 - \frac{1}{\lambda}\frac{1}{s}\sum_{i\in S}\|\nabla F_i(\mathbf{x}_i)\|^2.$$

*Proof.* Using the definition of $F_i$, we get

$$\nabla F_i(\mathbf{x}_i) = \nabla f_i(\mathbf{x}_i) + \nabla h_i^S(\mathbf{v}) + \lambda(\mathbf{x}_i - \mathbf{v}).$$

Hence,

$$\langle \nabla f_i(\mathbf{x}_i) + \nabla h_i^S(\bar{\mathbf{x}}_S), \mathbf{v} - \mathbf{x}_i\rangle = \lambda\|\mathbf{v} - \mathbf{x}_i\|^2 + \langle \nabla h_i^S(\bar{\mathbf{x}}_S) - \nabla h_i^S(\mathbf{v}), \mathbf{v} - \mathbf{x}_i\rangle + \langle \nabla F_i(\mathbf{x}_i), \mathbf{v} - \mathbf{x}_i\rangle.$$

Taking the average over $i$ on both sides of the first display, we have

$$\frac{1}{s}\sum_{i\in S}\nabla f_i(\mathbf{x}_i) = \lambda(\mathbf{v} - \bar{\mathbf{x}}_S) + \frac{1}{s}\sum_{i\in S}\nabla F_i(\mathbf{x}_i). \tag{B.13}$$

Therefore,

$$\frac{1}{2\lambda}\left\|\frac{1}{s}\sum_{i\in S}\nabla f_i(\mathbf{x}_i)\right\|^2 = \frac{1}{2\lambda}\left\|\lambda(\mathbf{v} - \bar{\mathbf{x}}_S) + \frac{1}{s}\sum_{i\in S}\nabla F_i(\mathbf{x}_i)\right\|^2$$

$$= \frac{\lambda}{2}\|\mathbf{v} - \bar{\mathbf{x}}_S\|^2 + \frac{1}{s}\sum_{i\in S}\langle \nabla F_i(\mathbf{x}_i), \mathbf{v} - \bar{\mathbf{x}}_S\rangle + \frac{1}{2\lambda}\left\|\frac{1}{s}\sum_{i\in S}\nabla F_i(\mathbf{x}_i)\right\|^2.$$

It follows that

$$\frac{1}{s}\sum_{i\in S}\langle \nabla f_i(\mathbf{x}_i) + \nabla h_i^S(\bar{\mathbf{x}}_S), \mathbf{v} - \mathbf{x}_i\rangle - \frac{1}{2\lambda}\left\|\frac{1}{s}\sum_{i\in S}\nabla f_i(\mathbf{x}_i)\right\|^2$$

$$= \lambda\frac{1}{s}\sum_{i\in S}\|\mathbf{v} - \mathbf{x}_i\|^2 - \frac{\lambda}{2}\|\mathbf{v} - \bar{\mathbf{x}}_S\|^2 + \frac{1}{s}\sum_{i\in S}\langle \nabla h_i^S(\bar{\mathbf{x}}_S) - \nabla h_i^S(\mathbf{v}), \mathbf{v} - \mathbf{x}_i\rangle$$

$$+ \frac{1}{s}\sum_{i\in S}\langle \nabla F_i(\mathbf{x}_i), \bar{\mathbf{x}}_S - \mathbf{x}_i\rangle - \frac{1}{2\lambda}\left\|\frac{1}{s}\sum_{i\in S}\nabla F_i(\mathbf{x}_i)\right\|^2$$

$$\stackrel{(B.8)}{=} \frac{\lambda}{2}\|\mathbf{v} - \bar{\mathbf{x}}_S\|^2 + \lambda\frac{1}{s}\sum_{i\in S}\|\mathbf{x}_i - \bar{\mathbf{x}}_S\|^2 + \frac{1}{s}\sum_{i\in S}\langle \nabla h_i^S(\bar{\mathbf{x}}_S) - \nabla h_i^S(\mathbf{v}), \bar{\mathbf{x}}_S - \mathbf{x}_i\rangle$$

$$+ \frac{1}{s}\sum_{i\in S}\langle \nabla F_i(\mathbf{x}_i), \bar{\mathbf{x}}_S - \mathbf{x}_i\rangle - \frac{1}{2\lambda}\left\|\frac{1}{s}\sum_{i\in S}\nabla F_i(\mathbf{x}_i)\right\|^2$$

$$\stackrel{(B.6)}{\geq} \frac{\lambda}{2}\|\mathbf{v} - \bar{\mathbf{x}}_S\|^2 + \lambda\frac{1}{s}\sum_{i\in S}\|\mathbf{x}_i - \bar{\mathbf{x}}_S\|^2 - \frac{1}{2\delta_s}\frac{1}{s}\sum_{i\in S}\|\nabla h_i^S(\bar{\mathbf{x}}_S) - \nabla h_i^S(\mathbf{v})\|^2 - \frac{\delta_s}{2}\frac{1}{s}\sum_{i\in S}\|\mathbf{x}_i - \bar{\mathbf{x}}_S\|^2$$

$$- \frac{\lambda}{2}\frac{1}{s}\sum_{i\in S}\|\mathbf{x}_i - \bar{\mathbf{x}}_S\|^2 - \frac{1}{2\lambda}\frac{1}{s}\sum_{i\in S}\|\nabla F_i(\mathbf{x}_i)\|^2 - \frac{1}{2\lambda}\left\|\frac{1}{s}\sum_{i\in S}\nabla F_i(\mathbf{x}_i)\right\|^2$$

$$\stackrel{(5),(B.8)}{\geq} \frac{\lambda - \delta_s}{2}\|\mathbf{v} - \bar{\mathbf{x}}_S\|^2 + \frac{\lambda - \delta_s}{2}\frac{1}{s}\sum_{i\in S}\|\mathbf{x}_i - \bar{\mathbf{x}}_S\|^2 - \frac{1}{\lambda}\frac{1}{s}\sum_{i\in S}\|\nabla F_i(\mathbf{x}_i)\|^2$$

$$\stackrel{(B.8)}{=} \frac{\lambda - \delta_s}{2}\frac{1}{s}\sum_{i\in S}\|\mathbf{v} - \mathbf{x}_i\|^2 - \frac{1}{\lambda}\frac{1}{s}\sum_{i\in S}\|\nabla F_i(\mathbf{x}_i)\|^2,$$

where in the second equality, we use the fact that $\frac{1}{s}\sum_{i\in S}[\nabla h_i^S(\bar{\mathbf{x}}_S) - \nabla h_i^S(\mathbf{v})] = 0$. $\qquad\square$

## C Proofs for S-DANE (Algorithm 1)

### C.1 One-Step Recurrence

**Lemma 16.** *Consider Algorithm 1. Let $f_i : \mathbb{R}^d \to \mathbb{R}$ be $\mu$-convex with $\mu \geq 0$ for any $i \in [n]$. Assume that $\{f_i\}_{i=1}^n$ have $\delta_s$-SOD. Then, for any $r \geq 0$, we have*

$$\frac{1}{\lambda}[f_{S_r}(\mathbf{x}^{r+1}) - f_{S_r}(\mathbf{x}^\star)] + \frac{1 + \mu/\lambda}{2}\|\mathbf{v}^{r+1} - \mathbf{x}^\star\|^2$$
$$\leq \frac{1}{2}\|\mathbf{v}^r - \mathbf{x}^\star\|^2 - \frac{1 - \delta_s/\lambda}{2}\frac{1}{s}\sum_{i\in S_r}\|\mathbf{v}^r - \mathbf{x}_{i,r+1}\|^2 + \frac{1}{\lambda^2}\frac{1}{s}\sum_{i\in S_r}\|\nabla F_{i,r}(\mathbf{x}_{i,r+1})\|^2.$$

*Proof.* By $\mu$-convexity of $f_i$, for any $r \geq 0$, it holds that

$$\frac{1}{\lambda}f_{S_r}(\mathbf{x}^\star) + \frac{1}{2}\|\mathbf{v}^r - \mathbf{x}^\star\|^2 = \frac{1}{\lambda}\frac{1}{s}\sum_{i\in S_r}f_i(\mathbf{x}^\star) + \frac{1}{2}\|\mathbf{v}^r - \mathbf{x}^\star\|^2$$

$$\stackrel{(B.1)}{\geq} \frac{1}{\lambda}\frac{1}{s}\sum_{i\in S_r}\left[f_i(\mathbf{x}_{i,r+1}) + \langle \nabla f_i(\mathbf{x}_{i,r+1}), \mathbf{x}^\star - \mathbf{x}_{i,r+1}\rangle + \frac{\mu}{2}\|\mathbf{x}_{i,r+1} - \mathbf{x}^\star\|^2\right] + \frac{1}{2}\|\mathbf{v}^r - \mathbf{x}^\star\|^2.$$

Recall that $\mathbf{v}^{r+1}$ is the minimizer of the final expression in $\mathbf{x}^\star$. This expression is a $(1 + \mu/\lambda)$-convex function in $\mathbf{x}^\star$. We can then estimate it by:

$$\frac{1}{\lambda}f_{S_r}(\mathbf{x}^\star) + \frac{1}{2}\|\mathbf{v}^r - \mathbf{x}^\star\|^2$$
$$\stackrel{(B.1)}{\geq} \frac{1}{\lambda}\frac{1}{s}\sum_{i\in S_r}\left[f_i(\mathbf{x}_{i,r+1}) + \langle \nabla f_i(\mathbf{x}_{i,r+1}), \mathbf{v}^{r+1} - \mathbf{x}_{i,r+1}\rangle + \frac{\mu}{2}\|\mathbf{x}_{i,r+1} - \mathbf{v}^{r+1}\|^2\right]$$
$$+ \frac{1}{2}\|\mathbf{v}^r - \mathbf{v}^{r+1}\|^2 + \frac{1 + \mu/\lambda}{2}\|\mathbf{v}^{r+1} - \mathbf{x}^\star\|^2.$$

Using the convexity of $f_i$ and dropping the non-negative $\frac{\mu}{2\lambda}\frac{1}{s}\sum_{i\in S_r}\|\mathbf{x}_{i,r+1} - \mathbf{v}^{r+1}\|^2$, we further get

$$\frac{1}{\lambda}f_{S_r}(\mathbf{x}^\star) + \frac{1}{2}\|\mathbf{v}^r - \mathbf{x}^\star\|^2$$
$$\stackrel{(B.1)}{\geq} \frac{1}{\lambda}\frac{1}{s}\sum_{i\in S_r}[f_i(\mathbf{x}^{r+1}) + \langle \nabla f_i(\mathbf{x}^{r+1}), \mathbf{x}_{i,r+1} - \mathbf{x}^{r+1}\rangle] + \frac{1 + \mu/\lambda}{2}\|\mathbf{v}^{r+1} - \mathbf{x}^\star\|^2$$
$$+ \frac{1}{\lambda}\frac{1}{s}\sum_{i\in S_r}\langle \nabla f_i(\mathbf{x}_{i,r+1}), \mathbf{v}^r - \mathbf{x}_{i,r+1}\rangle + \frac{1}{\lambda}\left\langle \frac{1}{s}\sum_{i\in S_r}\nabla f_i(\mathbf{x}_{i,r+1}), \mathbf{v}^{r+1} - \mathbf{v}^r\right\rangle$$
$$+ \frac{1}{2}\|\mathbf{v}^{r+1} - \mathbf{v}^r\|^2$$

$$\overset{\text{(B.6)}}{\geq} \frac{1}{\lambda}f_{S_r}(\mathbf{x}^{r+1}) + \frac{1+\mu/\lambda}{2}\|\mathbf{v}^{r+1} - \mathbf{x}^\star\|^2 + \frac{1}{\lambda}\frac{1}{s}\sum_{i\in S_r}\langle\nabla f_i(\mathbf{x}^{r+1}), \mathbf{x}_{i,r+1} - \mathbf{x}^{r+1}\rangle$$

$$+ \frac{1}{\lambda}\frac{1}{s}\sum_{i\in S_r}\langle\nabla f_i(\mathbf{x}_{i,r+1}), \mathbf{v}^r - \mathbf{x}_{i,r+1}\rangle - \frac{1}{2\lambda^2}\left\|\frac{1}{s}\sum_{i\in S_r}\nabla f_i(\mathbf{x}_{i,r+1})\right\|^2.$$

Denote $h_i^r := f_{S_r} - f_i$. Note that

$$\sum_{i\in S_r}\langle\nabla f_i(\mathbf{x}^{r+1}), \mathbf{x}_{i,r+1} - \mathbf{x}^{r+1}\rangle = \sum_{i\in S_r}\langle-\nabla h_i^r(\mathbf{x}^{r+1}), \mathbf{x}_{i,r+1} - \mathbf{v}^r\rangle,$$

where we have used:

$$\mathbf{x}^{r+1} = \frac{1}{s}\sum_{i\in S_r}\mathbf{x}_{i,r+1} \qquad\text{and}\qquad \sum_{i\in S^r}\nabla h_i^r(\mathbf{x}^{r+1}) = 0.$$

It follows that

$$\frac{1}{\lambda}f_{S_r}(\mathbf{x}^\star) + \frac{1}{2}\|\mathbf{v}^r - \mathbf{x}^\star\|^2 \geq \frac{1}{\lambda}f_{S_r}(\mathbf{x}^{r+1}) + \frac{1+\mu/\lambda}{2}\|\mathbf{v}^{r+1} - \mathbf{x}^\star\|^2$$

$$+ \frac{1}{\lambda}\frac{1}{s}\sum_{i\in S_r}\langle\nabla f_i(\mathbf{x}_{i,r+1}) + \nabla h_i^r(\mathbf{x}^{r+1}), \mathbf{v}^r - \mathbf{x}_{i,r+1}\rangle - \frac{1}{2\lambda^2}\left\|\frac{1}{s}\sum_{i\in S_r}\nabla f_i(\mathbf{x}_{i,r+1})\right\|^2.$$

We now apply Lemma 15 (with $\mathbf{x}_i = \mathbf{x}_{i,r+1}$, $\mathbf{v} = \mathbf{v}^r$, $S = S_r$ and $\mathbf{x} = \mathbf{x}^{r+1}$) to get

$$\frac{1}{s}\sum_{i\in S_r}\langle\nabla f_i(\mathbf{x}_{i,r+1}) + \nabla h_i^r(\mathbf{x}^{r+1}), \mathbf{v}^r - \mathbf{x}_{i,r+1}\rangle - \frac{1}{2\lambda}\left\|\frac{1}{s}\sum_{i\in S_r}\nabla f_i(\mathbf{x}_{i,r+1})\right\|^2$$

$$\geq \frac{\lambda - \delta_s}{2}\frac{1}{s}\sum_{i\in S_r}\|\mathbf{v}^r - \mathbf{x}_{i,r+1}\|^2 - \frac{1}{\lambda}\frac{1}{s}\sum_{i\in S_r}\|\nabla F_{i,r}(\mathbf{x}_{i,r+1})\|^2.$$

Substituting this lower bound into the previous display, we get the claim. $\qquad\square$

## C.2  Full Client Participation (Proof of Theorem 1)

*Proof.* Applying Lemma 16 and using $\sum_{i=1}^n\|\nabla F_{i,r}(\mathbf{x}_{i,r+1})\|^2 \leq \delta^2\sum_{i=1}^n\|\mathbf{x}_{i,r+1} - \mathbf{v}^r\|^2$ and $\lambda = 2\delta$, for any $r \geq 0$, we have

$$\frac{1}{\lambda}[f(\mathbf{x}^{r+1}) - f^\star] + \frac{1+\mu/\lambda}{2}\|\mathbf{v}^{r+1} - \mathbf{x}^\star\|^2$$

$$\leq \frac{1}{2}\|\mathbf{v}^r - \mathbf{x}^\star\|^2 - \frac{1-\delta/\lambda}{2}\frac{1}{n}\sum_{i=1}^n\|\mathbf{v}^r - \mathbf{x}_{i,r+1}\|^2 + \frac{1}{\lambda^2}\frac{1}{n}\sum_{i=1}^n\|\nabla F_{i,r}(\mathbf{x}_{i,r+1})\|^2$$

$$\leq \frac{1}{2}\|\mathbf{v}^r - \mathbf{x}^\star\|^2 - \Big(\frac{1-1/2}{2} - \frac{1}{4}\Big)\|\mathbf{v}^r - \mathbf{x}_{i,r+1}\|^2 = \frac{1}{2}\|\mathbf{v}^r - \mathbf{x}^\star\|^2.$$

Rearranging, we get

$$\frac{2q}{\lambda}[f(\mathbf{x}^{r+1}) - f^\star] \leq q\|\mathbf{v}^r - \mathbf{x}^\star\|^2 - \|\mathbf{v}^{r+1} - \mathbf{x}^\star\|^2.$$

where $q := \frac{1}{1+\mu/\lambda}$. Applying Lemma 11 with $\varepsilon = 0$ and using convexity of $f$, we obtain

$$\frac{2q}{\lambda}[f(\bar{\mathbf{x}}^R) - f^\star] + \frac{1-q}{1-q^R}\|\mathbf{v}^R - \mathbf{x}^\star\|^2 \leq \frac{1-q}{(1/q)^R - 1}\|\mathbf{v}^0 - \mathbf{x}^\star\|^2 = \frac{1-q}{(1/q)^R - 1}D^2.$$

Dropping the non-negative term $\frac{1-q}{1-q^R}\|\mathbf{v}^R - \mathbf{x}^\star\|^2$ and rearranging, we get

$$f(\bar{\mathbf{x}}^R) - f^\star \leq \frac{(1-q)\lambda}{2q\big[(1/q)^R - 1\big]}D^2.$$

Plugging in the choice of $\lambda$ and the definition of $q$, we get the claim. $\qquad\square$

**Corollary 17.** *Under the same setting as in Theorem 1, to achieve $f(\bar{\mathbf{x}}^R) - f^\star \leq \varepsilon$, we need at most the following number of communication rounds:*

$$R = \mathcal{O}\Big(\frac{\mu+\delta}{\mu}\log\Big(1 + \frac{\mu D^2}{\varepsilon}\Big)\Big).$$

*Proof.* Using the fact that $(1+q)^k \geq \exp(\frac{q}{1+q}k)$ for any $q \geq 0$ and $k > 0$, we get

$$f(\bar{\mathbf{x}}^R) - f^\star \leq \frac{\mu D^2}{2[(1 + \frac{\mu}{2\delta})^R - 1]} \leq \frac{\mu D^2}{2[\exp(\frac{\mu}{\mu+2\delta}R) - 1]} \leq \varepsilon.$$

Rearranging, we get the claim. $\qquad\square$

**Proof of Corollary 3.**

*Proof.* To achieve $\sum_{i=1}^{n}\|\nabla F_{i,r}(\mathbf{x}_{i,r+1})\|^2 \leq \delta^2 \sum_{i=1}^{n}\|\mathbf{x}_{i,r+1} - \mathbf{y}^r\|^2$, for each $i \in [n]$, it is sufficient to ensure that $\|\nabla F_{i,r}(\mathbf{x}_{i,r+1})\| \leq \delta\|\mathbf{x}_{i,r+1} - \mathbf{v}^r\|$. Let $\mathbf{x}_{i,r}^{\star} \coloneqq \arg\min_{\mathbf{x}} F_{i,r}(\mathbf{x})$. Since

$$\|\mathbf{x}_{i,r+1} - \mathbf{v}^r\| \geq \|\mathbf{v}^r - \mathbf{x}_{i,r}^{\star}\| - \|\mathbf{x}_{i,r+1} - \mathbf{x}_{i,r}^{\star}\| \overset{\text{(B.2)}}{\geq} \|\mathbf{v}^r - \mathbf{x}_{i,r}^{\star}\| - \frac{1}{\lambda}\|\nabla F_{i,r}(\mathbf{x}_{i,r+1})\|$$

and $\lambda = 2\delta$, it suffices to ensure that

$$\|\nabla F_{i,r}(\mathbf{x}_{i,r+1})\| \leq \frac{2\delta}{3}\|\mathbf{v}^r - \mathbf{x}_{i,r}^{\star}\| \tag{C.1}$$

for any $i \in [n]$. According to Theorem 2 from [33] (or Theorem 3.2 from [37]), there exists a certain algorithm such that when started from the point $\mathbf{v}^r$, after $K$ queries to $\nabla F_{i,r}$, it generates the point $\mathbf{v}_{i,r+1}$ such that

$$\|\nabla F_{i,r}(\mathbf{x}_{i,r+1})\| \leq \mathcal{O}\left(\frac{(L+\lambda)\|\mathbf{v}^r - \mathbf{x}_{i,r}^{\star}\|}{K^2}\right) = \mathcal{O}\left(\frac{L\|\mathbf{v}^r - \mathbf{x}_{i,r}^{\star}\|}{K^2}\right)$$

(recall that $\delta \leq L$). Setting $K = \Theta(\sqrt{\frac{L}{\delta}})$ concludes the proof. $\qquad\square$

**Remark 18.** *Recall that $F_{i,r}$ is $(L + \lambda)$-smooth and $(\mu + \lambda)$-convex, $\lambda = \Theta(\delta)$ and $\delta \leq L$. Suppose worker $i$ uses the standard* GD *to approximately solve the local subproblem at round $r$ starting at $\mathbf{v}^r$ for $K$ steps and return the last point, then by Lemma 19, we have that $\|\nabla F_{i,r}(\mathbf{x}_{i,r+1})\|^2 \leq \mathcal{O}\left(\frac{(L+\lambda)^2\|\mathbf{v}^r - \mathbf{x}_{i,r}^{\star}\|^2}{K^2}\right)$. To satisfy the accuracy condition (C.1), it is sufficient to make $K = \Theta(\frac{L}{\delta})$ local steps. Suppose worker $i$ uses the fast gradient method, then by Theorem 3.18 from [4], we have that $\|\nabla F_{i,r}(\mathbf{x}_{i,r+1})\|^2 \leq 2(L + \lambda)\big(F_{i,r}(\mathbf{x}_{i,r+1}) - F_{i,r}(\mathbf{x}_{i,r}^{\star})\big) \leq \mathcal{O}\big((L + \lambda)^2 \exp(-\sqrt{\frac{\mu+\lambda}{L+\lambda}}K)\|\mathbf{v}^r - \mathbf{x}_{i,r}^{\star}\|^2\big)$. To satisfy the accuracy condition (C.1), it suffices to make $K = \Theta(\sqrt{\frac{L+\delta}{\mu+\delta}}\log(\frac{L+\delta}{\delta})) = \Theta(\sqrt{\frac{L}{\mu+\delta}}\log(\frac{L}{\delta}))$ gradient oracle calls.*

**Lemma 19** (Theorem 2.2.5 in [49]). *Let $f : \mathbb{R}^d \to \mathbb{R}$ be a convex and $L$-smooth function. Consider the gradient method with constant stepsize:*

$$\mathbf{x}_{k+1} = \mathbf{x}_k - \frac{1}{L}\nabla f(\mathbf{x}_k), \qquad k \geq 0,$$

*started from some $\mathbf{x}_0 \in \mathbb{R}^d$. Then, for any $K \geq 1$, it holds that*

$$\|\nabla f(\mathbf{x}_K)\| \leq \mathcal{O}\left(\frac{L\|\mathbf{x}_0 - \mathbf{x}^{\star}\|}{K}\right). \tag{C.2}$$

*Proof.* By Theorem 2.2.5 in [49], we have that

$$\min_{k \in [K]}\|\nabla f(\mathbf{x}_k)\| \leq \mathcal{O}\left(\frac{L\|\mathbf{x}_0 - \mathbf{x}^{\star}\|}{K}\right).$$

It remains to note that the algorithm generates non-increasing $\|\nabla f(\mathbf{x}_k)\|$ since

$$\begin{aligned}
\|\nabla f(\mathbf{x}_{k+1})\|^2 &= \|\nabla f(\mathbf{x}_{k+1}) - \nabla f(\mathbf{x}_k) + \nabla f(\mathbf{x}_k)\|^2 \\
&= \|\nabla f(\mathbf{x}_{k+1}) - \nabla f(\mathbf{x}_k)\|^2 + 2\langle\nabla f(\mathbf{x}_{k+1}) - \nabla f(\mathbf{x}_k), \nabla f(\mathbf{x}_k)\rangle + \|\nabla f(\mathbf{x}_k)\|^2 \\
&= \|\nabla f(\mathbf{x}_{k+1}) - \nabla f(\mathbf{x}_k)\|^2 - 2L\langle\nabla f(\mathbf{x}_k) - \nabla f(\mathbf{x}_{k+1}), \mathbf{x}_k - \mathbf{x}_{k+1}\rangle + \|\nabla f(\mathbf{x}_k)\|^2 \\
&\overset{\text{(B.5)}}{\leq} \|\nabla f(\mathbf{x}_k)\|^2 - \|\nabla f(\mathbf{x}_{k+1}) - \nabla f(\mathbf{x}_k)\|^2 \leq \|\nabla f(\mathbf{x}_k)\|^2. \qquad\square
\end{aligned}$$

## C.3 Partial Client Participation (Proof of Theorem 5)

The following theorem is a slight extension of Theorem 5, which includes the use of stochastic local solvers.

**Theorem 20.** *Consider Algorithm 1. Let $f_i : \mathbb{R}^d \to \mathbb{R}$ be $\mu$-convex with $\mu \geq 0$ for any $i \in [n]$ and let $n \geq 2$. Assume that $\{f_i\}_{i=1}^n$ have $\delta_s$-SOD, $\Delta_s$-ED and $\zeta$-BGV. Let $\lambda = \frac{4(n-s)}{s(n-1)}\frac{\zeta^2}{\varepsilon} + 2(\delta_s + \Delta_s)$. For any $r \geq 0$, suppose we have*

$$\frac{1}{s}\sum_{i \in S_r}\mathbb{E}_{\xi_{i,r}}[\|\nabla F_{i,r}(\mathbf{x}_{i,r+1})\|^2] \leq \frac{\lambda^2}{4}\frac{1}{s}\sum_{i \in S_r}\mathbb{E}_{\xi_{i,r}}[\|\mathbf{x}_{i,r+1} - \mathbf{v}^r\|^2] + \frac{\lambda\varepsilon}{4}, \tag{C.3}$$

*for some $\varepsilon > 0$, where $\xi_{i,r}$ denotes the randomness coming from device $i$ when solving its subproblem at round $r$. We assume that $\{\xi_{i,r}\}$ are independent random variables. To reach $\mathbb{E}[f(\bar{\mathbf{x}}^R) - f^{\star}] \leq \varepsilon$, we need at most the following number of communication rounds:*

$$R = \Theta\left(\left[\frac{\delta_s + \Delta_s + \mu}{\mu} + \frac{n-s}{n-1}\frac{\zeta^2}{s\varepsilon\mu}\right]\log\left(1 + \frac{\mu D^2}{\varepsilon}\right)\right) \leq \Theta\left(\frac{(\delta_s + \Delta_s)D^2}{\varepsilon} + \frac{n-s}{n-1}\frac{\zeta^2 D^2}{s\varepsilon^2}\right),$$

*where $\bar{\mathbf{x}}^R \coloneqq \sum_{r=1}^{R}p^r\mathbf{x}^r / \sum_{r=1}^{R}p^r$, $p \coloneqq 1 + \frac{\mu}{\lambda}$, and $D \coloneqq \|\mathbf{x}^0 - \mathbf{x}^{\star}\|$.*

*Proof.* According to Lemma 16, we have for any $r \geq 0$,

$$\frac{1}{\lambda}[f_{S_r}(\mathbf{x}^{r+1}) - f_{S_r}(\mathbf{x}^\star)] + \frac{1+\mu/\lambda}{2}\|\mathbf{v}^{r+1} - \mathbf{x}^\star\|^2$$
$$\leq \frac{1}{2}\|\mathbf{v}^r - \mathbf{x}^\star\|^2 - \frac{1-\delta_s/\lambda}{2}\frac{1}{s}\sum_{i \in S_r}\|\mathbf{v}^r - \mathbf{x}_{i,r+1}\|^2 + \frac{1}{\lambda^2}\frac{1}{s}\sum_{i \in S_r}\|\nabla F_{i,r}(\mathbf{x}_{i,r+1})\|^2.$$

According to Lemma 14 (with $S = S_r$, $\mathbf{x} = \mathbf{x}^{r+1}$ and $\mathbf{y} = \mathbf{v}^r$), for any $\gamma > 0$, we have

$$\mathbb{E}_{S_r}[f(\mathbf{x}^{r+1}) - f_{S_r}(\mathbf{x}^{r+1})] \leq \frac{n-s}{n-1}\frac{\gamma\zeta^2}{2s} + \left(\frac{1}{2\gamma} + \frac{\Delta_s}{2}\right)\mathbb{E}_{S_r}[\|\mathbf{x}^{r+1} - \mathbf{v}^r\|^2]$$
$$\overset{\text{(B.8)}}{\leq} \frac{n-s}{n-1}\frac{\gamma\zeta^2}{2s} + \left(\frac{1}{2\gamma} + \frac{\Delta_s}{2}\right)\mathbb{E}_{S_r}\left[\frac{1}{s}\sum_{i \in S_r}\|\mathbf{x}_{i,r+1} - \mathbf{v}^r\|^2\right].$$

Adding $\frac{1}{\lambda}f(\mathbf{x}^{r+1})$ to both sides of the first display, taking the expectation over $S_r$ on both sides, substituting the previous upper bound and setting $\gamma = \frac{s(n-1)\varepsilon}{2\zeta^2(n-s)}$, we get

$$\frac{1}{\lambda}\mathbb{E}_{S_r}[f(\mathbf{x}^{r+1}) - f^\star] + \frac{1+\mu/\lambda}{2}\mathbb{E}_{S_r}[\|\mathbf{v}^{r+1} - \mathbf{x}^\star\|^2]$$
$$\leq \frac{1}{2}\|\mathbf{v}^r - \mathbf{x}^\star\|^2 - \left(\frac{1}{2} - \frac{\delta_s + \Delta_s}{2\lambda} - \frac{1}{2\gamma\lambda}\right)\mathbb{E}_{S_r}\left[\frac{1}{s}\sum_{i \in S_r}\|\mathbf{x}_{i,r+1} - \mathbf{v}^r\|^2\right]$$
$$+ \frac{\varepsilon}{4\lambda} + \frac{1}{\lambda^2}\mathbb{E}_{S_r}\left[\frac{1}{s}\sum_{i \in S_r}\|\nabla F_{i,r}(\mathbf{x}_{i,r+1})\|^2\right].$$

Denote all the randomness $\{\xi_{i,r}\}_{i \in S_r}$ by $\xi_r$. Since $\xi_{i,r}$ is independent of the choice of $S_r$ for any $i \in [n]$, taking the expectation over $\xi_r$ on both sides of the previous display and using our assumption (C.3), we obtain

$$\frac{1}{\lambda}\mathbb{E}_{S_r,\xi_r}[f(\mathbf{x}^{r+1}) - f^\star] + \frac{1+\mu/\lambda}{2}\mathbb{E}_{S_r,\xi_r}[\|\mathbf{v}^{r+1} - \mathbf{x}^\star\|^2]$$
$$\leq \frac{1}{2}\|\mathbf{v}^r - \mathbf{x}^\star\|^2 - \left(\frac{1}{4} - \frac{\delta_s + \Delta_s}{2\lambda} - \frac{1}{2\gamma\lambda}\right)\mathbb{E}_{S_r,\xi_r}\left[\frac{1}{s}\sum_{i \in S_r}\|\mathbf{x}_{i,r+1} - \mathbf{v}^r\|^2\right] + \frac{\varepsilon}{2\lambda}.$$

By our choice of $\lambda$, we have $\frac{1}{4} - \frac{\delta_s + \Delta_s}{2} - \frac{1}{2\gamma} \geq 0$. Taking the full expectation on both sides, we get

$$\frac{1}{\lambda}\mathbb{E}[f(\mathbf{x}^{r+1}) - f^\star] + \frac{1+\mu/\lambda}{2}\mathbb{E}[\|\mathbf{v}^{r+1} - \mathbf{x}^\star\|^2] \leq \frac{1}{2}\mathbb{E}[\|\mathbf{v}^r - \mathbf{x}^\star\|^2] + \frac{\varepsilon}{2\lambda}.$$

According to Lemma 11 and the fact that $\|\mathbf{v}^0 - \mathbf{x}^\star\| = D$, we get

$$\frac{2}{\mu+\lambda}\mathbb{E}[f(\bar{\mathbf{x}}^R) - f^\star] + (1-q)\mathbb{E}[\|\mathbf{v}^R - \mathbf{x}^\star\|^2] \leq \frac{1-q}{(1/q)^R - 1}D^2 + \frac{1}{\mu+\lambda}\varepsilon,$$

where $q := \frac{1}{1+\mu/\lambda}$. Rearranging and dropping the non-negative $\mathbb{E}[\|\mathbf{v}^R - \mathbf{x}^\star\|^2]$, we get, for any $R \geq 1$,

$$\mathbb{E}[f(\bar{\mathbf{x}}^R) - f^\star] \leq \frac{\mu D^2}{2[(\frac{\mu}{\lambda}+1)^R - 1]} + \frac{\varepsilon}{2} \leq \frac{\mu D^2}{2[\exp(\frac{\mu}{\mu+\lambda}R) - 1]} + \frac{\varepsilon}{2}.$$

To reach $\varepsilon$-accuracy, it suffices to let $\frac{\mu D^2}{2[\exp(\frac{\mu}{\mu+\lambda}R) - 1]} \leq \frac{\varepsilon}{2}$. Rearranging gives the claim. $\qquad\square$

### C.3.1 Stochastic Local Solver

Note that there exist many stochastic optimization algorithms that can also achieve the accuracy condition (C.3) such as variance reduction methods [23, 10], adaptive SGD methods [13], etc. Here, we take the simplest algorithm: SGD with constant stepsize as an example.

**Corollary 21.** *Consider Algorithm 1 under the same settings as in Theorem 20. Further assume that each $f_i$ is $L$-smooth and each device has access to mini-batch stochastic gradient $g_i(\mathbf{x}, \bar{\xi}_i)$ such that*

$$\mathbb{E}_{\bar{\xi}_i}[g_i(\mathbf{x}, \bar{\xi}_i)] = \nabla f_i(\mathbf{x}), \qquad \mathbb{E}_{\bar{\xi}_i}[\|g_i(\mathbf{x}, \bar{\xi}_i) - \nabla f_i(\mathbf{x})\|^2] \leq \sigma^2.$$

*Suppose for any $r \geq 0$, each device $i \in S_r$ solves its subproblem approximately by using mini-batch SGD:*

$$\mathbf{z}_{k+1} = \mathbf{z}_k - \frac{1}{H}\left[g_i(\mathbf{x}, \bar{\xi}_{i,k}^r) - \nabla f_i(\mathbf{v}^r) + \nabla f_{S_r}(\mathbf{v}^r) + \lambda(\mathbf{z}_k - \mathbf{v}^r)\right], \qquad 0 \leq k \leq K,$$

where $\mathbf{z}_0 = \mathbf{v}^r$ and $H > L + \lambda$ is the stepsize coefficient. Let $\xi_{i,r}$ denote $(\bar{\xi}_{i,k}^r)_k$. To achieve accuracy condition (C.3) for an appropriately chosen $H$, each device $i$ requires at most the following number of stochastic mini-batch oracle calls:

$$K = \Theta\left(\left[\frac{L+\lambda}{\mu+\lambda} + \frac{(L+\lambda)\sigma^2}{(\mu+\lambda)\lambda\varepsilon}\right]\log\frac{L+\lambda}{\lambda}\right).$$

*Proof.* To get (C.3), it suffices to ensure that, for any $i \in S_r$, we have

$$E_{\xi_{i,r}}[\|\nabla F_{i,r}(\mathbf{x}_{i,r+1})\|^2] \leq \frac{\lambda^2}{4}\,\mathbb{E}_{\xi_{i,r}}[\|\mathbf{x}_{i,r+1} - \mathbf{v}^r\|^2] + \frac{\lambda\varepsilon}{4}.$$

For this, it suffices to ensure that

$$\mathbb{E}_{\xi_{i,r}}[\|\nabla F_{i,r}(\mathbf{x}_{i,r+1})\|^2] \leq \frac{\lambda^2}{10}\|\mathbf{v}^r - \mathbf{x}_{i,r}^\star\|^2 + \frac{\lambda\varepsilon}{5}. \tag{C.4}$$

where $\mathbf{x}_{i,r}^\star := \arg\min_{\mathbf{x}} F_{i,r}(\mathbf{x})$. Indeed, suppose (C.4) holds, then we have

$$\|\mathbf{x}_{i,r+1} - \mathbf{v}^r\| \geq \|\mathbf{v}^r - \mathbf{x}_{i,r}^\star\| - \|\mathbf{x}_{i,r+1} - \mathbf{x}_{i,r}^\star\| \overset{(\mathrm{B.2})}{\geq} \|\mathbf{v}^r - \mathbf{x}_{i,r}^\star\| - \frac{1}{\lambda}\|\nabla F_{i,r}(\mathbf{x}_{i,r+1})\|.$$

Hence,

$$\|\mathbf{v}^r - \mathbf{x}_{i,r}^\star\|^2 \leq \frac{2}{\lambda^2}\|\nabla F_{i,r}(\mathbf{x}_{i,r+1})\|^2 + 2\|\mathbf{x}_{i,r+1} - \mathbf{v}^r\|^2.$$

Plugging in this inequality into (C.4) and taking expectation w.r.t $\xi_{i,r}$ on both sides, we get

$$\mathbb{E}_{\xi_{i,r}}[\|\nabla F_{i,r}(\mathbf{x}_{i,r+1})\|^2] \leq \frac{1}{5}\,\mathbb{E}_{\xi_{i,r}}[\|\nabla F_{i,r}(\mathbf{x}_{i,r+1})\|^2] + \frac{\lambda^2}{5}\,\mathbb{E}_{\xi_{i,r}}[\|\mathbf{x}_{i,r+1} - \mathbf{v}^r\|^2] + \frac{\lambda}{5}\varepsilon.$$

Rearranging gives the weaker condition.

We next consider the number of mini-batch stochastic gradient oracles required for SGD to achieve (C.4). Since $F_{i,r}$ is $(L+\lambda)$-smooth and $(\mu+\lambda)$-convex, according to Lemma 22, we have

$$\mathbb{E}_{\xi_{i,r}}[\|\nabla F_{i,r}(\bar{\mathbf{z}}_K)\|^2] \leq 2(L+\lambda)\,\mathbb{E}_{\xi_{i,r}}[F_{i,r}(\bar{\mathbf{z}}_K) - F_{i,r}^\star]$$

$$\leq 2(L+\lambda)\left(\frac{(\mu+\lambda)\|\mathbf{v}^r - \mathbf{x}_{i,r}^\star\|^2}{2[\exp((\mu+\lambda)K/H) - 1]} + \frac{\sigma^2}{2(H - L - \lambda)}\right),$$

where $\bar{\mathbf{z}}_K := \frac{1}{\sum_{k=1}^K \frac{1}{q^k}}\sum_{k=1}^K \frac{\mathbf{z}_k}{q^k}$ and $q = \frac{H-\mu-\lambda}{H}$. Choosing now $H = (L+\lambda) + \frac{5(L+\lambda)\sigma^2}{\lambda\varepsilon}$, and letting the coefficient of the first part in the previous display be $\leq \frac{\lambda^2}{10}$, we get the claim. $\qquad\square$

**Lemma 22.** *Let $f$ be a $\mu$-convex and $L$-smooth function. Consider SGD with constant stepsize $H > L$:*

$$\mathbf{x}_{k+1} := \arg\min_{\mathbf{x}\in\mathbb{R}^d}\left\{\langle g_k, \mathbf{x}\rangle + \frac{H}{2}\|\mathbf{x} - \mathbf{x}_k\|^2\right\},$$

*where $g_k := g(\mathbf{x}_k, \xi_k)$ with $\mathbb{E}_\xi[g(\mathbf{x}, \xi)] = \nabla f(\mathbf{x})$ and $\mathbb{E}_\xi[\|g(\mathbf{x}, \xi) - \nabla f(\mathbf{x})\|^2] \leq \sigma^2$ for any $\mathbf{x} \in \mathbb{R}^d$. Then for any $K \geq 1$, we have*

$$\mathbb{E}[f(\bar{\mathbf{x}}_K)] - f^\star \leq \frac{\mu\|\mathbf{x}_0 - \mathbf{x}^\star\|^2}{2[\exp(\mu K/H) - 1]} + \frac{\sigma^2}{2(H - L)}. \tag{C.5}$$

*where $\bar{\mathbf{x}}_K := \frac{1}{\sum_{k=1}^K \frac{1}{q^k}}\sum_{k=1}^K \frac{\mathbf{x}_k}{q^k}$ and $q = \frac{H-\mu}{H}$.*

*Proof.* Indeed, for any $k \geq 0$, we have

$$f(\mathbf{x}_k) + \langle g_k, \mathbf{x}^\star - \mathbf{x}_k\rangle + \frac{H}{2}\|\mathbf{x}_k - \mathbf{x}^\star\|^2$$

$$\geq f(\mathbf{x}_k) + \langle g_k, \mathbf{x}_{k+1} - \mathbf{x}_k\rangle + \frac{H}{2}\|\mathbf{x}_{k+1} - \mathbf{x}_k\|^2 + \frac{H}{2}\|\mathbf{x}_{k+1} - \mathbf{x}^\star\|^2$$

$$\overset{(\mathrm{B.3})}{\geq} f(\mathbf{x}_{k+1}) + \langle g_k - \nabla f(\mathbf{x}_k), \mathbf{x}_{k+1} - \mathbf{x}_k\rangle + \frac{H - L}{2}\|\mathbf{x}_{k+1} - \mathbf{x}_k\|^2 + \frac{H}{2}\|\mathbf{x}_{k+1} - \mathbf{x}^\star\|^2$$

$$\overset{(\mathrm{B.6})}{\geq} f(\mathbf{x}_{k+1}) - \frac{\|g_k - \nabla f(\mathbf{x}_k)\|^2}{2(H - L)} + \frac{H}{2}\|\mathbf{x}_{k+1} - \mathbf{x}^\star\|^2.$$

Taking the expectation on both sides and using $\mu$-convexity of $f$, we get

$$\mathbb{E}[f(\mathbf{x}_{k+1}) - f^\star] + \frac{H}{2}\,\mathbb{E}[\|\mathbf{x}_{k+1} - \mathbf{x}^\star\|^2] \leq \frac{H - \mu}{2}\,\mathbb{E}[\|\mathbf{x}_k - \mathbf{x}^\star\|^2] + \frac{\sigma^2}{2(H - L)}.$$

Applying Lemma 11, we have for any $K \geq 1$:

$$\mathbb{E}[f(\bar{\mathbf{x}}_K) - f^\star] \leq \frac{\mu\|\mathbf{x}_0 - \mathbf{x}^\star\|^2}{2[(1/q)^K - 1]} + \frac{\sigma^2}{2(H - L)} \leq \frac{\mu\|\mathbf{x}_0 - \mathbf{x}^\star\|^2}{2[\exp(\mu K/H) - 1]} + \frac{\sigma^2}{2(H - L)}. \qquad\square$$

# D Proofs for Accelerated S-DANE (Algorithm 2)

## D.1 One-Step Recurrence

**Lemma 23.** *Consider Algorithm 2. Let $f_i : \mathbb{R}^d \to \mathbb{R}$ be $\mu$-convex with $\mu \geq 0$ for any $i \in [n]$. Assume that $\{f_i\}_{i=1}^n$ have $\delta_s$-SOD. For any $r \geq 0$, we have*

$$A_r f_{S_r}(\mathbf{x}^r) + a_{r+1} f_{S_r}(\mathbf{x}^\star) + \frac{B_r}{2} \|\mathbf{v}^r - \mathbf{x}^\star\|^2$$

$$\geq A_{r+1} f_{S_r}(\mathbf{x}^{r+1}) + \frac{B_{r+1}}{2} \|\mathbf{v}^{r+1} - \mathbf{x}^\star\|^2$$

$$+ A_{r+1} \left( \frac{\lambda - \delta_s}{2} \frac{1}{s} \sum_{i \in S_r} \|\mathbf{x}_{i,r+1} - \mathbf{y}^r\|^2 - \frac{1}{\lambda} \frac{1}{s} \sum_{i \in S_r} \|\nabla F_{i,r}(\mathbf{x}_{i,r+1})\|^2 \right).$$

*Proof.* By $\mu$-convexity of $f_i$, for any $r \geq 0$, it holds that

$$A_r f_{S_r}(\mathbf{x}^r) + a_{r+1} f_{S_r}(\mathbf{x}^\star) + \frac{B_r}{2} \|\mathbf{v}^r - \mathbf{x}^\star\|^2$$

$$= A_r \frac{1}{s} \sum_{i \in S_r} f_i(\mathbf{x}^r) + a_{r+1} \frac{1}{s} \sum_{i \in S_r} f_i(\mathbf{x}^\star) + \frac{B_r}{2} \|\mathbf{v}^r - \mathbf{x}^\star\|^2$$

$$\overset{\text{(B.1)}}{\geq} A_r \frac{1}{s} \sum_{i \in S_r} [f_i(\mathbf{x}_{i,r+1}) + \langle \nabla f_i(\mathbf{x}_{i,r+1}), \mathbf{x}^r - \mathbf{x}_{i,r+1} \rangle] + \frac{B_r}{2} \|\mathbf{v}^r - \mathbf{x}^\star\|^2$$

$$+ a_{r+1} \frac{1}{s} \sum_{i \in S_r} \left[ f_i(\mathbf{x}_{i,r+1}) + \langle \nabla f_i(\mathbf{x}_{i,r+1}), \mathbf{x}^\star - \mathbf{x}_{i,r+1} \rangle + \frac{\mu}{2} \|\mathbf{x}_{i,r+1} - \mathbf{x}^\star\|^2 \right].$$

Recall that $\mathbf{v}^{r+1}$ is the minimizer of the final expression in $\mathbf{x}^\star$. This expression is a $(\mu a_{r+1} + B_r)$-convex function in $\mathbf{x}^\star$. By convexity and using the fact that $A_{r+1} = A_r + a_{r+1}$ and $B_{r+1} = \mu a_{r+1} + B_r$, we obtain

$$A_r f_{S_r}(\mathbf{x}^r) + a_{r+1} f_{S_r}(\mathbf{x}^\star) + \frac{B_r}{2} \|\mathbf{v}^r - \mathbf{x}^\star\|^2$$

$$\geq A_{r+1} \frac{1}{s} \sum_{i \in S_r} f_i(\mathbf{x}_{i,r+1}) + \frac{\mu a_{r+1}}{2} \frac{1}{s} \sum_{i \in S_r} \|\mathbf{x}_{i,r+1} - \mathbf{v}^{r+1}\|^2 + \frac{B_r}{2} \|\mathbf{v}^r - \mathbf{v}^{r+1}\|^2$$

$$+ \frac{1}{s} \sum_{i \in S_r} \langle \nabla f_i(\mathbf{x}_{i,r+1}), A_r \mathbf{x}^r + a_{r+1} \mathbf{v}^{r+1} - A_{r+1} \mathbf{x}_{i,r+1} \rangle + \frac{B_{r+1}}{2} \|\mathbf{v}^{r+1} - \mathbf{x}^\star\|^2.$$

Recall that $\mathbf{y}^r = \frac{A_r}{A_{r+1}} \mathbf{x}^r + \frac{a_{r+1}}{A_{r+1}} \mathbf{v}^r$. Therefore,

$$\frac{B_r}{2} \|\mathbf{v}^r - \mathbf{v}^{r+1}\|^2 + \frac{1}{s} \sum_{i \in S_r} \langle \nabla f_i(\mathbf{x}_{i,r+1}), A_r \mathbf{x}^r + a_{r+1} \mathbf{v}^{r+1} - A_{r+1} \mathbf{x}_{i,r+1} \rangle$$

$$= \frac{B_r}{2} \|\mathbf{v}^r - \mathbf{v}^{r+1}\|^2 + a_{r+1} \left\langle \frac{1}{s} \sum_{i \in S_r} \nabla f_i(\mathbf{x}_{i,r+1}), \mathbf{v}^{r+1} - \mathbf{v}^r \right\rangle$$

$$+ A_{r+1} \frac{1}{s} \sum_{i \in S_r} \langle \nabla f_i(\mathbf{x}_{i,r+1}), \mathbf{y}^r - \mathbf{x}_{i,r+1} \rangle$$

$$\overset{\text{(B.6)}}{\geq} - \frac{a_{r+1}^2}{2 B_r} \left\| \frac{1}{s} \sum_{i \in S_r} \nabla f_i(\mathbf{x}_{i,r+1}) \right\|^2 + A_{r+1} \frac{1}{s} \sum_{i \in S_r} \langle \nabla f_i(\mathbf{x}_{i,r+1}), \mathbf{y}^r - \mathbf{x}_{i,r+1} \rangle.$$

Substituting this lower bound, using convexity of $f_i$ and dropping the non-negative $\frac{\mu a_{r+1}}{2} \frac{1}{s} \sum_{i \in S_r} \|\mathbf{x}_{i,r+1} - \mathbf{v}^{r+1}\|^2$, we further get

$$A_r f_{S_r}(\mathbf{x}^r) + a_{r+1} f_{S_r}(\mathbf{x}^\star) + \frac{B_r}{2} \|\mathbf{v}^r - \mathbf{x}^\star\|^2$$

$$\overset{\text{(B.1)}}{\geq} A_{r+1} \frac{1}{s} \sum_{i \in S_r} [f_i(\mathbf{x}^{r+1}) + \langle \nabla f_i(\mathbf{x}^{r+1}), \mathbf{x}_{i,r+1} - \mathbf{x}^{r+1} \rangle] + \frac{B_{r+1}}{2} \|\mathbf{v}^{r+1} - \mathbf{x}^\star\|^2$$

$$+ A_{r+1} \frac{1}{s} \sum_{i \in S_r} \langle \nabla f_i(\mathbf{x}_{i,r+1}), \mathbf{y}^r - \mathbf{x}_{i,r+1} \rangle - \frac{a_{r+1}^2}{2 B_r} \left\| \frac{1}{s} \sum_{i \in S_r} \nabla f_i(\mathbf{x}_{i,r+1}) \right\|^2.$$

Denote $h_i^r := f_{S_r} - f_i$. Substituting

$$\sum_{i \in S_r} \langle \nabla f_i(\mathbf{x}^{r+1}), \mathbf{x}_{i,r+1} - \mathbf{x}^{r+1} \rangle = \sum_{i \in S_r} \langle -\nabla h_i^r(\mathbf{x}^{r+1}), \mathbf{x}_{i,r+1} - \mathbf{y}^r \rangle$$

into the previous display, we get

$$A_r f_{S_r}(\mathbf{x}^r) + a_{r+1} f_{S_r}(\mathbf{x}^\star) + \frac{B_r}{2} \|\mathbf{v}^r - \mathbf{x}^\star\|^2$$

$$\geq A_{r+1} f_{S_r}(\mathbf{x}^{r+1}) + \frac{B_{r+1}}{2} \|\mathbf{v}^{r+1} - \mathbf{x}^\star\|^2$$

$$+ A_{r+1} \frac{1}{s} \sum_{i \in S_r} \langle \nabla f_i(\mathbf{x}_{i,r+1}) + \nabla h_i^r(\mathbf{x}^{r+1}), \mathbf{y}^r - \mathbf{x}_{i,r+1} \rangle - \frac{a_{r+1}^2}{2B_r} \left\| \frac{1}{s} \sum_{i \in S_r} \nabla f_i(\mathbf{x}_{i,r+1}) \right\|^2.$$

We now apply Lemma 15 (with $\mathbf{x}_i = \mathbf{x}_{i,r+1}$, $\mathbf{v} = \mathbf{y}^r$, $S = S_r$ and $\mathbf{x} = \mathbf{x}^{r+1}$) to get

$$\frac{1}{s} \sum_{i \in S_r} \langle \nabla f_i(\mathbf{x}_{i,r+1}) + \nabla h_i^r(\mathbf{x}^{r+1}), \mathbf{y}^r - \mathbf{x}_{i,r+1} \rangle - \frac{1}{2\lambda} \left\| \frac{1}{s} \sum_{i \in S_r} \nabla f_i(\mathbf{x}_{i,r+1}) \right\|^2$$

$$\geq \frac{\lambda - \delta_s}{2} \frac{1}{s} \sum_{i \in S_r} \|\mathbf{y}^r - \mathbf{x}_{i,r+1}\|^2 - \frac{1}{\lambda} \frac{1}{s} \sum_{i \in S_r} \|\nabla F_{i,r}(\mathbf{x}_{i,r+1})\|^2.$$

Substituting this lower bound into the previous display and using $A_{r+1} = \frac{a_{r+1}^2 \lambda}{B_r}$, we get the claim. $\qquad \square$

## D.2 Full Client Participation (Proof of Theorem 6)

*Proof.* Applying Lemma 23 and using $\sum_{i=1}^n \|\nabla F_{i,r}(\mathbf{x}_{i,r+1})\|^2 \leq \delta^2 \sum_{i=1}^n \|\mathbf{x}_{i,r+1} - \mathbf{y}^r\|^2$ and $\lambda = 2\delta$, for any $r \geq 0$, we have

$$A_r f(\mathbf{x}^r) + a_{r+1} f^\star + \frac{B_r}{2} \|\mathbf{v}^r - \mathbf{x}^\star\|^2$$

$$\geq A_{r+1} f(\mathbf{x}^{r+1}) + \frac{B_{r+1}}{2} \|\mathbf{v}^{r+1} - \mathbf{x}^\star\|^2 + A_{r+1} \left( \frac{\lambda - \delta}{2} - \frac{\delta^2}{\lambda} \right) \frac{1}{s} \sum_{i \in S_r} \|\mathbf{x}_{i,r+1} - \mathbf{y}^r\|^2$$

$$= A_{r+1} f(\mathbf{x}^{r+1}) + \frac{B_{r+1}}{2} \|\mathbf{v}^{r+1} - \mathbf{x}^\star\|^2.$$

Subtracting $A_{r+1} f^\star$ on both sides, we get

$$A_{r+1}[f(\mathbf{x}^{r+1}) - f^\star] + \frac{B_{r+1}}{2} \|\mathbf{v}^{r+1} - \mathbf{x}^\star\|^2 \leq A_r[f(\mathbf{x}^r) - f^\star] + \frac{B_r}{2} \|\mathbf{v}^r - \mathbf{x}^\star\|^2.$$

Recursively applying the previous display from $r = 0$ to $r = R - 1$, we get

$$A_R[f(\mathbf{x}^R) - f^\star] + \frac{B_R}{2} \|\mathbf{v}^R - \mathbf{x}^\star\|^2 \leq A_0[f(\mathbf{x}^0) - f^\star] + \frac{1}{2} \|\mathbf{v}^0 - \mathbf{x}^\star\|^2 = \frac{1}{2} \|\mathbf{x}^0 - \mathbf{x}^\star\|^2.$$

It remains to apply Lemma 12 and plug in the estimation of the growth of $A_R$. $\qquad \square$

**Corollary 24.** *Under the same setting as in Theorem 6, to achieve $f(\mathbf{x}^R) - f^\star \leq \varepsilon$, we need at most the following number of communication rounds:*

$$R = \mathcal{O}\left( \sqrt{\frac{\delta + \mu}{\mu}} \log\left( 1 + \sqrt{\frac{\min\{\mu, \delta\} D^2}{\varepsilon}} \right) \right).$$

*Proof.* When $\mu \leq 8\delta$, by using

$$\left( 1 + \sqrt{\frac{\mu}{8\delta}} \right)^R - \left( 1 - \sqrt{\frac{\mu}{8\delta}} \right)^R \geq \left( 1 + \sqrt{\frac{\mu}{8\delta}} \right)^R - 1 \geq \exp\left( \frac{\sqrt{\mu} R}{\sqrt{8\delta} + \sqrt{\mu}} \right) - 1,$$

we get

$$f(\mathbf{x}^R) - f^\star \leq \frac{2\mu D^2}{\left[ \left( 1 + \sqrt{\frac{\mu}{8\delta}} \right)^R - \left( 1 - \sqrt{\frac{\mu}{8\delta}} \right)^R \right]^2} \leq \frac{2\mu D^2}{\left[ \exp\left( \frac{\sqrt{\mu} R}{\sqrt{8\delta} + \sqrt{\mu}} \right) - 1 \right]^2}.$$

Making the right-hand side $\leq \varepsilon$ and rearranging, we get the claim. When $\mu \geq 8\delta$, it suffices to ensure that $\frac{\delta D^2}{4^{R-2}} \leq \varepsilon$. $\qquad \square$

### D.3 Partial Client Participation

It is well known that accelerated stochastic gradient methods are not able to improve the complexity in the stochastic part compared with the basic methods [11]. A similar result is also shown for our accelerated distributed method.

**Theorem 25.** *Consider Algorithm 2 under the same setting as in Theorem 20. Let*

$$\lambda = \Theta\bigg( (\delta_s + \Delta_s) + \frac{(n-s)R}{s(n-1)}\frac{\zeta^2}{\varepsilon} \bigg)$$

*and suppose that, for any $r \geq 0$, we have*

$$\frac{1}{s}\sum_{i \in S_r} \mathbb{E}_{\xi_{i,r}}[\|\nabla F_{i,r}(\mathbf{x}_{i,r+1})\|^2] \leq \mathcal{O}\bigg( \frac{\lambda^2}{4}\frac{1}{s}\sum_{i \in S_r} \mathbb{E}_{\xi_{i,r}}[\|\mathbf{x}_{i,r+1} - \mathbf{v}^r\|^2] + \frac{\lambda\varepsilon}{4R} \bigg).$$

*Denote $D := \|\mathbf{x}^0 - \mathbf{x}^\star\|$. Then, to ensure that $\mathbb{E}[f(\mathbf{x}^R)] - f^\star \leq \varepsilon$ for some $\varepsilon > 0$, we need to perform at most the following number of communication rounds:*

$$R = \Theta\bigg( \frac{\sqrt{\delta_s + \Delta_s} + \sqrt{\mu}}{\sqrt{\mu}}\log\bigg(1 + \sqrt{\frac{\min\{\mu,\lambda\}D^2}{\varepsilon}}\bigg) + \frac{n-s}{n-1}\frac{\zeta^2}{s\varepsilon\mu}\log^2\bigg(1 + \sqrt{\frac{\min\{\mu,\lambda\}D^2}{\varepsilon}}\bigg)\bigg)$$

$$\leq \Theta\bigg( \sqrt{\frac{(\delta_s + \Delta_s)D^2}{\varepsilon}} + \frac{n-s}{n-1}\frac{\zeta^2 D^2}{s\varepsilon^2} \bigg).$$

The error term that depends on $\zeta^2$ and $\varepsilon$ is at the same scale as S-DANE, i.e. $\mathcal{O}(\frac{\zeta^2}{\varepsilon})$ when $\mu > 0$ and $\mathcal{O}(\frac{\zeta^2}{\varepsilon^2})$ when $\mu = 0$. Nevertheless, when $s$ is large enough such that this second error becomes no larger than the first optimization term, then ACC-S-DANE can still be faster than S-DANE.

**Proof of Theorem 25.**

*Proof.* According to Lemma 23, we have, for any $r \geq 0$,

$$A_r f_{S_r}(\mathbf{x}^r) + a_{r+1}f_{S_r}(\mathbf{x}^\star) + \frac{B_r}{2}\|\mathbf{v}^r - \mathbf{x}^\star\|^2$$

$$\geq A_{r+1}f_{S_r}(\mathbf{x}^{r+1}) + \frac{B_{r+1}}{2}\|\mathbf{v}^{r+1} - \mathbf{x}^\star\|^2$$

$$+ A_{r+1}\frac{\lambda - \delta_s}{2}\frac{1}{s}\sum_{i \in S_r}\|\mathbf{x}_{i,r+1} - \mathbf{y}^r\|^2 - A_{r+1}\frac{1}{\lambda}\frac{1}{s}\sum_{i \in S_r}\|\nabla F_{i,r}(\mathbf{x}_{i,r+1})\|^2.$$

According to Lemma 14 (with $S = S_r$, $\mathbf{x} = \mathbf{x}^{r+1}$ and $\mathbf{y} = \mathbf{y}^r$), for any $\gamma > 0$, we have

$$\mathbb{E}_{S_r}\big[f(\mathbf{x}^{r+1}) - f_{S_r}(\mathbf{x}^{r+1})\big] \leq \frac{n-s}{n-1}\frac{\gamma\zeta^2}{2s} + \bigg(\frac{1}{2\gamma} + \frac{\Delta_s}{2}\bigg)\mathbb{E}_{S_r}[\|\mathbf{x}^{r+1} - \mathbf{y}^r\|^2]$$

$$\overset{(B.8)}{\leq} \frac{n-s}{n-1}\frac{\gamma\zeta^2}{2s} + \bigg(\frac{1}{2\gamma} + \frac{\Delta_s}{2}\bigg)\mathbb{E}_{S_r}\bigg[\frac{1}{s}\sum_{i \in S_r}\|\mathbf{x}_{i,r+1} - \mathbf{y}^r\|^2\bigg].$$

Adding $A_{r+1}f(\mathbf{x}^{r+1})$ to both sides of the first display, taking the expectation over $S_r$ on both sides, substituting the previous upper bound, and setting $\gamma = \frac{s(n-1)\varepsilon'}{2\zeta^2(n-s)}$ with $\varepsilon' > 0$, we get

$$A_r f(\mathbf{x}^r) + a_{r+1}f^\star + \frac{B_r}{2}\|\mathbf{v}^r - \mathbf{x}^\star\|^2$$

$$\geq A_{r+1}\mathbb{E}_{S_r}[f(\mathbf{x}^{r+1})] + \frac{B_{r+1}}{2}\mathbb{E}_{S_r}[\|\mathbf{v}^{r+1} - \mathbf{x}^\star\|^2]$$

$$+ A_{r+1}\bigg(\frac{\lambda}{2} - \frac{\delta_s + \Delta_s}{2} - \frac{1}{2\gamma}\bigg)\mathbb{E}_{S_r}\bigg[\frac{1}{s}\sum_{i \in S_r}\|\mathbf{x}_{i,r+1} - \mathbf{y}^r\|^2\bigg]$$

$$- \frac{A_{r+1}}{4}\varepsilon' - \frac{A_{r+1}}{\lambda}\mathbb{E}_{S_r}\bigg[\frac{1}{s}\sum_{i \in S_r}\|\nabla F_{i,r}(\mathbf{x}_{i,r+1})\|^2\bigg].$$

Denote all the randomness $\{\xi_{i,r}\}_{i \in S_r}$ by $\xi_r$. Since $\xi_{i,r}$ is independent of the choice of $S_r$ for any $i \in [n]$, taking the expectation over $\xi_r$ on both sides of the previous display and using the assumption that $\mathbb{E}_{S_r,\xi_r}\big[\frac{1}{s}\sum_{i \in S_r}\|\nabla F_{i,r}(\mathbf{x}_{i,r+1})\|^2\big] \leq \mathbb{E}_{S_r,\xi_r}\big[\frac{\lambda^2}{4}\frac{1}{s}\sum_{i \in S_r}\|\mathbf{x}_{i,r+1} - \mathbf{y}^r\|^2\big] + \frac{\lambda\varepsilon'}{4}$, we obtain

$$A_r f(\mathbf{x}^r) + a_{r+1}f^\star + \frac{B_r}{2}\|\mathbf{v}^r - \mathbf{x}^\star\|^2$$

$$\geq A_{r+1}\,\mathbb{E}_{S_r,\xi_r}[f(\mathbf{x}^{r+1})] + \frac{B_{r+1}}{2}\,\mathbb{E}_{S_r,\xi_r}[\|\mathbf{v}^{r+1} - \mathbf{x}^\star\|^2]$$

$$+ A_{r+1}\Big(\frac{\lambda}{4} - \frac{\delta_s + \Delta_s}{2} - \frac{1}{2\gamma}\Big)\mathbb{E}_{S_r,\xi_r}\Big[\frac{1}{s}\sum_{i\in S_r}\|\mathbf{x}_{i,r+1} - \mathbf{y}^r\|^2\Big] - \frac{A_{r+1}}{2}\varepsilon'.$$

By choosing $\lambda = \frac{4\zeta^2(n-s)}{s(n-1)\varepsilon'} + 2(\delta_s + \Delta_s)$, we have that $\frac{\lambda}{4} - \frac{(\delta_s+\Delta_s)}{2} - \frac{1}{2\gamma} \geq 0$. Taking the full expectation on both sides, we get

$$A_r\,\mathbb{E}[f(\mathbf{x}^r)] + a_{r+1}f^\star + \frac{B_r}{2}\,\mathbb{E}[\|\mathbf{v}^r - \mathbf{x}^\star\|^2] \geq A_{r+1}\,\mathbb{E}[f(\mathbf{x}^{r+1})] + \frac{B_{r+1}}{2}\,\mathbb{E}[\|\mathbf{v}^{r+1} - \mathbf{x}^\star\|^2] - \frac{A_{r+1}}{2}\varepsilon'.$$

Subtracting $A_{r+1}f(\mathbf{x}^\star)$ on both sides, summing up from $r = 0$ to $r = R - 1$ and using the fact that $A_0 = 0$, $\mathbf{v}_0 = \mathbf{x}_0$ and $B_0 = 1$, we get

$$A_R\,\mathbb{E}[f(\mathbf{x}^R) - f^\star] + \frac{B_R}{2}\,\mathbb{E}[\|\mathbf{v}^R - \mathbf{x}^\star\|^2] \leq \frac{1}{2}\|\mathbf{x}^0 - \mathbf{x}^\star\|^2 + \frac{\varepsilon'}{2}\sum_{r=1}^{R}A_r.$$

Dividing both sides by $A_R$, setting $\varepsilon' = \frac{\varepsilon}{R}$ and using the fact that the sequence $\{A_r\}$ is non-decreasing, we get

$$\mathbb{E}[f(\mathbf{x}^R)] - f^\star \leq \frac{1}{2A_R}\|\mathbf{x}^0 - \mathbf{x}^\star\|^2 + \frac{\varepsilon}{2}.$$

We now apply Lemma 12 with $c = \lambda$ to get

$$A_R \geq \frac{[(1 + \sqrt{\frac{\mu}{4\lambda}})^R - (1 - \sqrt{\frac{\mu}{4\lambda}})^R]^2}{4\mu} \geq \frac{[(1 + \sqrt{\frac{\mu}{4\lambda}})^R - 1]^2}{4\mu} \geq \frac{[\exp(\frac{\sqrt{\mu}R}{\sqrt{4\lambda}+\sqrt{\mu}}) - 1]^2}{4\mu}$$

when $\mu \leq 4\lambda$, and $A_R \geq \frac{1}{4\lambda}(1 + \sqrt{\frac{\mu}{4\lambda}})^{2(R-1)}$ when $\mu \geq 4\lambda$. Letting these lower bounds be larger than $\frac{\|\mathbf{x}^0-\mathbf{x}^\star\|^2}{\varepsilon}$, we get

$$R = \Omega\Bigg(\frac{\sqrt{\mu} + \sqrt{\lambda}}{\sqrt{\mu}}\log\bigg(1 + \sqrt{\frac{\min\{\mu,\lambda\}\|\mathbf{x}_0 - \mathbf{x}^\star\|^2}{\varepsilon}}\bigg)\Bigg).$$

Plugging $\lambda = \Omega\big(\frac{\zeta^2(n-s)R}{sn\varepsilon} + (\delta_s + \Delta_s)\big)$ into the last display and rearranging, we get the condition for $R$. $\quad\square$

### D.3.1 Stochastic Local Solver

**Corollary 26.** *Consider Algorithm 2 under the same settings as in Theorem 25. Consider the same stochastic local solver used in Corollary 21. To achieve*

$$\frac{1}{s}\sum_{i\in S_r}\mathbb{E}_{\xi_{i,r}}[\|\nabla F_{i,r}(\mathbf{x}_{i,r+1})\|^2] \leq \mathcal{O}\bigg(\frac{\lambda^2}{4}\frac{1}{s}\sum_{i\in S_r}\mathbb{E}_{\xi_{i,r}}[\|\mathbf{x}_{i,r+1} - \mathbf{y}^r\|^2] + \frac{\lambda\varepsilon}{4R}\bigg),$$

*each device $i$ requires at most the following number of stochastic mini-batch oracle calls:*

$$K = \Theta\bigg(\Big[\frac{L + \lambda}{\mu + \lambda} + \frac{(L + \lambda)\sigma^2 R}{(\mu + \lambda)\lambda\varepsilon}\Big]\log\frac{L + \lambda}{\lambda}\bigg).$$

*Proof.* The proof is the same as that for Corollary 21. $\quad\square$

## E  Dynamic Estimation of Similarity Constant by Line Search

**Theorem 27.** *Consider Algorithm 3. Suppose that each function $f_i$ is $\mu$-convex for some $\mu \geq 0$, and $\{f_i\}_{i=1}^n$ have $\delta$-SOD for some $\delta > 0$. Let $\tilde{\lambda} \leq 2\delta$. Then, for any $R \geq 1$, it holds that*

$$f(\bar{\mathbf{x}}^R) - f^\star \leq \frac{\mu D^2}{2[(1 + \frac{\mu}{4\delta})^R - 1]} \leq \frac{2\delta D^2}{R},$$

*where $\bar{\mathbf{x}}^R := \arg\min_{\mathbf{x}\in\{\mathbf{x}^1,\dots,\mathbf{x}^R\}} f(\mathbf{x})$. To ensure that $f(\bar{\mathbf{x}}^R) - f^\star \leq \varepsilon$ for any given $\varepsilon > 0$, it suffices to set*

$$R = \Theta\bigg(\frac{\delta + \mu}{\mu}\log\Big(1 + \frac{\mu D^2}{\varepsilon}\Big)\bigg),$$

*where $D := \|\mathbf{x}^0 - \mathbf{x}^\star\|$. Furthermore, the total number of communication rounds spent inside the $r$- and $k$-loops since the start of the algorithm and up to the moment $\bar{\mathbf{x}}^R$ has been computed is*

$$\mathcal{O}(1)\sum_{k=0}^{R-1}(k_r + 1) \leq \mathcal{O}\Big(R + \log\frac{2\delta}{\tilde{\lambda}}\Big).$$

---

**Algorithm 3** S-DANE with line search

---

1: **Input:** $\tilde{\lambda} > 0, \mu \geq 0, \mathbf{x}^0 = \mathbf{v}^0 \in \mathbb{R}^d$. Let $h_i := f - f_i$.
2: Set $\lambda_{0,0} = \tilde{\lambda}$.
3: **for** $r = 0, 1, 2, \ldots$ **do**
4:     **for** $k = 0, 1, \ldots$ **do**
5:         **for each device** $i \in [n]$ **in parallel do**
6:             $\mathbf{x}_{i,r+1,k} \approx \arg\min_{\mathbf{x} \in \mathbb{R}^d} \left\{ F_{i,r,k}(\mathbf{x}) := f_i(\mathbf{x}) + \langle \nabla h_i(\mathbf{v}^r), \mathbf{x} \rangle + \frac{\lambda_{r,k}}{2} \|\mathbf{x} - \mathbf{v}^r\|^2 \right\}$.
7:             (stop running the local solver once $\|\nabla F_{i,r,k}(\mathbf{x}_{i,r+1,k})\| \leq \frac{\lambda_{r,k}}{2} \|\mathbf{x}_{i,r+1,k} - \mathbf{v}^r\|$)
8:         Aggregate local models: $\mathbf{x}^{r+1,k} = \frac{1}{n} \sum_{i=1}^{n} \mathbf{x}_{i,r+1,k}$.
9:         **if** $\frac{1}{n} \sum_{i=1}^{n} \langle \nabla f_i(\mathbf{x}_{i,r+1,k}) + \nabla h_i(\mathbf{x}^{r+1,k}), \mathbf{v}^r - \mathbf{x}_{i,r+1,k} \rangle \geq \frac{1}{2\lambda_{r,k}} \left\| \frac{1}{n} \sum_{i=1}^{n} \nabla f_i(\mathbf{x}_{i,r+1,k}) \right\|^2$ **then**
10:             $k_r = k$ and **break** the loop.
11:         $\lambda_{r,k+1} = 2\lambda_{r,k}$.
12:     $\lambda_r = \lambda_{r,k_r}$, $\mathbf{x}_{i,r+1} = \mathbf{x}_{i,r+1,k_r}$, $\mathbf{x}^{r+1} = \mathbf{x}^{r+1,k_r}$, $\lambda_{r+1,0} = \frac{1}{2}\lambda_r$.
13:     $\mathbf{v}^{r+1} = \arg\min_{\mathbf{x} \in \mathbb{R}^d} \left\{ \frac{1}{n} \sum_{i=1}^{n} [\langle \nabla f_i(\mathbf{x}_{i,r+1}), \mathbf{x} \rangle + \frac{\mu}{2} \|\mathbf{x} - \mathbf{x}_{i,r+1}\|^2] + \frac{\lambda_r}{2} \|\mathbf{x} - \mathbf{v}^r\|^2 \right\}$.

---

*Proof.* According to Lemma 15 (with $\mathbf{x}_i = \mathbf{x}_{i,r+1}$, $\mathbf{v} = \mathbf{v}^r$, $S = [n]$ and $\bar{\mathbf{x}}_S = \mathbf{x}^{r+1}$) and our requirement on $\|\nabla F_{i,r}(\mathbf{x}_{i,r+1})\|$, whenever $\lambda_{r,k} \geq 2\delta$, we can estimate

$$\frac{1}{n} \sum_{i=1}^{n} \langle \nabla f_i(\mathbf{x}_{i,r+1}) + \nabla h_i(\mathbf{x}^{r+1}), \mathbf{v}^r - \mathbf{x}_{i,r+1} \rangle - \frac{1}{2\lambda_{r,k}} \left\| \frac{1}{n} \sum_{i=1}^{n} \nabla f_i(\mathbf{x}_{i,r+1}) \right\|^2$$

$$\geq \frac{\lambda_{r,k} - \delta}{2} \frac{1}{n} \sum_{i=1}^{n} \|\mathbf{v}^r - \mathbf{x}_{i,r+1}\|^2 - \frac{1}{\lambda_{r,k}} \frac{1}{n} \sum_{i=1}^{n} \|\nabla F_{i,r}(\mathbf{x}_{i,r+1})\|^2$$

$$\geq \frac{\lambda_{r,k} - 2\delta}{4} \frac{1}{n} \sum_{i=1}^{n} \|\mathbf{v}^r - \mathbf{x}_{i,r+1}\|^2 \geq 0.$$

Hence, at any iteration of the $r$-loop, the corresponding $k$-loop eventually terminates. Further, since $\lambda_{0,0} \leq 2\delta$, we can easily prove by induction that the quantities $\lambda_{r,k}$ stay reasonably bounded:

$$\lambda_{r,0} \leq 2\delta, \quad \lambda_r \equiv \lambda_{r,k_r} \leq 4\delta =: \lambda_{\max}, \qquad \forall r \geq 0. \tag{E.1}$$

Proceeding exactly in the same way as in the proof of Lemma 16 and using the termination condition of the $k$-loop, we conclude that, for any $r \geq 0$, it holds that

$$\frac{1}{\lambda_r}[f(\mathbf{x}^{r+1}) - f^\star] + \frac{1 + \mu/\lambda_r}{2} \|\mathbf{v}^{r+1} - \mathbf{x}^\star\|^2 \leq \frac{1}{2} \|\mathbf{v}^r - \mathbf{x}^\star\|^2.$$

In view of (E.1), this means that, for any $r \geq 0$,

$$\frac{1}{\lambda_{\max}}[f(\mathbf{x}^{r+1}) - f^\star] + \frac{1 + \mu/\lambda_{\max}}{2} \|\mathbf{v}^{r+1} - \mathbf{x}^\star\|^2 \leq \frac{1}{2} \|\mathbf{v}^r - \mathbf{x}^\star\|^2.$$

Following the same proof as in Appendix C.2 but with $\lambda$ replaced by $\lambda_{\max}$, we obtain the first two claims.

It remains to estimate the total number of communication rounds required to construct the point $\bar{\mathbf{x}}^R$. In order to carry out the $k$-loop, the server needs to compute $\nabla f(\mathbf{v}^r)$ and send this vector, as well as $\mathbf{v}^r$ and $\lambda_{r,0}$, to each client, which requires $\mathcal{O}(1)$ communication rounds. Every iteration of the $k$-loop also requires $\mathcal{O}(1)$ communication rounds, and the total number of such iterations is $k_r$. Thus, every iteration of the $r$-loop requires $\mathcal{O}(k_r + 1)$ communication rounds. Furthermore, during the corresponding rounds, the server may also additionally compute the function value $f(\mathbf{x}^{r+1})$ needed for updating the output point $\bar{\mathbf{x}}^{r+1}$; this could be done, e.g., inside the $k$-loop, alongside with the computation of the gradient $\nabla f(\mathbf{x}^{r+1,k})$ needed to evaluate the "if" condition. Thus, $\bar{\mathbf{x}}^R$ can be indeed computed after $\mathcal{O}(1) \sum_{r=0}^{R-1}(k_r + 1)$ communication rounds. To estimate the latter sum, observe that, by construction, for any $r \geq 0$, we have $\lambda_{r+1,0} \equiv \frac{1}{2}\lambda_{r,k_r} = 2^{k_r-1}\lambda_{r,0}$. Taking logarithms, we see that $k_r = 1 + \log_2 \frac{\lambda_{r+1,0}}{\lambda_{r,0}}$ for any $r \geq 0$. Thus,

$$\sum_{r=0}^{R-1}(k_r + 1) = \sum_{r=0}^{R-1}\left(2 + \log_2 \frac{\lambda_{r+1,0}}{\lambda_{r,0}}\right) = 2R + \log_2 \frac{\lambda_{R,0}}{\lambda_{0,0}} \leq 2R + \log_2 \frac{2\delta}{\tilde{\lambda}},$$

where the final inequality is due to (E.1) and our choice of $\lambda_{0,0}$. $\qquad\square$

**Algorithm 4** Acc-S-DANE with line search

1: **Input:** $\tilde{\lambda} > 0$, $\mu \geq 0$, $\mathbf{x}^0 = \mathbf{v}^0 \in \mathbb{R}^d$. Let $h_i = f - f_i$.
2: Set $A_0 = 0$, $B_0 = 1$, $\lambda_{0,0} = \tilde{\lambda}$.
3: **for** $r = 0, 1, 2, \ldots$ **do**
4:      **for** $k = 0, 1, \ldots$ **do**
5:          Find $a_{r+1,k} > 0$ from the equation $\lambda_{r,k} = \frac{(A_r + a_{r+1,k})B_r}{a_{r+1,k}^2}$. Set $A_{r+1,k} = A_r + a_{r+1,k}$.
6:          $\mathbf{y}^{r,k} = \frac{A_r}{A_{r+1,k}}\mathbf{x}^r + \frac{a_{r+1,k}}{A_{r+1,k}}\mathbf{v}^r$.
7:          **for each device** $i \in [n]$ **in parallel do**
8:             $\mathbf{x}_{i,r+1,k} \approx \arg\min_{\mathbf{x} \in \mathbb{R}^d} \big\{ F_{i,r,k}(\mathbf{x}) := f_i(\mathbf{x}) + \langle \nabla h_i(\mathbf{y}^{r,k}), \mathbf{x} \rangle + \frac{\lambda_{r,k}}{2}\|\mathbf{x} - \mathbf{y}^{r,k}\|^2 \big\}$.
9:             (stop running the local solver once $\|\nabla F_{i,r,k}(\mathbf{x}_{i,r+1,k})\| \leq \frac{\lambda_{r,k}}{2}\|\mathbf{x}_{i,r+1,k} - \mathbf{y}^{r,k}\|$)
10:          Aggregate local models: $\mathbf{x}^{r+1,k} = \frac{1}{n}\sum_{i=1}^n \mathbf{x}_{i,r+1,k}$.
11:          **if** $\frac{1}{n}\sum_{i=1}^n \langle \nabla f_i(\mathbf{x}_{i,r+1,k}) + \nabla h_i(\mathbf{x}^{r+1,k}), \mathbf{y}^{r,k} - \mathbf{x}_{i,r+1,k} \rangle \geq \frac{1}{2\lambda_{r,k}}\big\|\frac{1}{n}\sum_{i=1}^n \nabla f_i(\mathbf{x}_{i,r+1,k})\big\|^2$ **then**
12:             $k_r = k$ and **break** the loop.
13:          $\lambda_{r,k+1} = 2\lambda_{r,k}$.
14:      $\lambda_r = \lambda_{r,k_r}$, $\mathbf{x}_{i,r+1} = \mathbf{x}_{i,r+1,k_r}$, $\mathbf{x}^{r+1} = \mathbf{x}^{r+1,k_r}$, $a_{r+1} = a_{r+1,k_r}$, $\lambda_{r+1,0} = \frac{1}{2}\lambda_r$.
15:      $A_{r+1} = A_r + a_{r+1}$, $B_{r+1} = B_r + \mu a_{r+1}$.
16:      $\mathbf{v}^{r+1} = \arg\min_{\mathbf{x} \in \mathbb{R}^d} \big\{ \frac{a_{r+1}}{n}\sum_{i=1}^n [\langle \nabla f_i(\mathbf{x}_{i,r+1}), \mathbf{x} \rangle + \frac{\mu}{2}\|\mathbf{x} - \mathbf{x}_{i,r+1}\|^2] + \frac{B_r}{2}\|\mathbf{x} - \mathbf{v}^r\|^2 \big\}$.

**Theorem 28.** *Consider Algorithm 4. Suppose that each function $f_i$ is $\mu$-convex for some $\mu \geq 0$, and $\{f_i\}_{i=1}^n$ have $\delta$-SOD for some $\delta > 0$. Let $\tilde{\lambda} \leq 2\delta$. If $\mu \leq 16\delta$, then, for any $R \geq 1$, it holds that*

$$f(\mathbf{x}^R) - f^\star \leq \frac{2\mu D^2}{\big[(1 + \sqrt{\frac{\mu}{16\delta}})^R - (1 - \sqrt{\frac{\mu}{16\delta}})^R\big]^2} \leq \frac{8\delta D^2}{R^2},$$

*where $D := \|\mathbf{x}^0 - \mathbf{x}^\star\|$. Otherwise, $f(\mathbf{x}^R) - f^\star \leq \frac{8\delta D^2}{(1+\sqrt{\frac{\mu}{16\delta}})^{2(R-1)}}$ for any $R \geq 1$. To ensure that $f(\mathbf{x}^R) - f^\star \leq \varepsilon$ for any given $\varepsilon > 0$, it suffices to set*

$$R = \Theta\bigg(\sqrt{\frac{\delta + \mu}{\mu}}\log\bigg(1 + \sqrt{\frac{\min\{\mu,\delta\}D^2}{\varepsilon}}\bigg)\bigg).$$

*Furthermore, the total number of communication rounds spent inside the $r$- and $k$-loops since the start of the algorithm and up to the moment $\mathbf{x}^R$ has been computed is*

$$\mathcal{O}(1)\sum_{k=0}^{R-1}(k_r + 1) \leq \mathcal{O}\bigg(R + \log\frac{2\delta}{\tilde{\lambda}}\bigg).$$

*Proof.* Using the same reasoning as in the proof of Theorem 27, we can justify that, at any iteration of the $r$-loop, the corresponding $k$-loop eventually terminates, and $\lambda_{r,k}$ stays uniformly bounded as in (E.1). Next, we proceed in the same way as in the proof of Lemma 23 and use the termination condition of the $k$-loop to obtain that, for any $r \geq 0$,

$$A_{r+1}[f(\mathbf{x}^{r+1}) - f^\star] + \frac{B_{r+1}}{2}\|\mathbf{v}^{r+1} - \mathbf{x}^\star\|^2 \leq A_r[f(\mathbf{x}^r) - f^\star] + \frac{B_r}{2}\|\mathbf{v}^r - \mathbf{x}^\star\|^2.$$

This shows that, for any $R \geq 1$,

$$f(\mathbf{x}^R) - f^\star \leq \frac{D^2}{2A_R}.$$

To estimate the rate of growth of $A_R$, we use the equation for $a_{r+1} \equiv a_{r+1,k_r}$ and the bound on $\lambda_r$ from (E.1). This gives us, for any $r \geq 0$, the following inequality:

$$\frac{A_{r+1}B_r}{a_{r+1}^2} = \lambda_r \leq \lambda_{\max},$$

where $B_r \equiv 1 + \mu A_r$. Invoking Lemma 12, we get a lower bound on $A_R$, and the first claim follows.

The bound on $R$ via $\varepsilon$ can be justified by the same argument as in the proof of Corollary 24.

To estimate the total number of communication rounds, we can follow exactly the same argument as in the proof of Theorem 27. $\qquad\square$

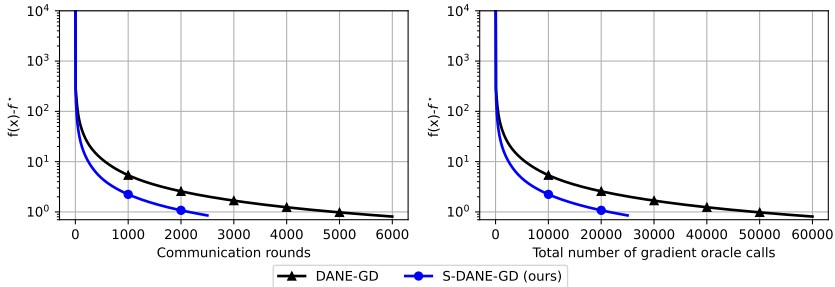

Figure E.1: Comparison of S-DANE against DANE for solving a convex quadratic minimization problem with the same number of local steps.

---

**Algorithm 5** S-DANE (DL)

---

1: **Input:** $\lambda > 0, \eta > 0, \gamma \in [0, 1], \mathbf{x}^0 = \mathbf{v}^0 \in \mathbb{R}^d, s \in [n]$
2: **for** $r = 0, 1, 2 \ldots$ **do**
3:     Sample $S_r \in \binom{[n]}{s}$ uniformly at random without replacement
4:     **for each device** $i \in S_r$ **in parallel do**
5:         Set $\mathbf{x}_{i,r+1} \approx \arg\min_{\mathbf{x} \in \mathbb{R}^d} \{F_{i,r}(\mathbf{x})\}$, where
6:
$$F_{i,r}(\mathbf{x}) := f_i(\mathbf{x}) - \langle \mathbf{x}, \nabla f_i(\mathbf{v}^r) - \nabla f_{S_r}(\mathbf{v}^r) \rangle + \frac{\lambda}{2} \|\mathbf{x} - \mathbf{v}^r\|^2. \quad \text{(option 1)}$$

7:
$$F_{i,r}(\mathbf{x}) := f_i(\mathbf{x}) + \frac{\lambda}{2} \|\mathbf{x} - \mathbf{v}^r\|^2. \qquad \text{(option 2)}$$

8:     Set $\mathbf{x}^{r+1} = \frac{1}{s} \sum_{i \in S_r} \mathbf{x}_{i,r+1}$
9:     Set $\mathbf{v}^{r+1} = \gamma \mathbf{x}^{r+1} + (1 - \gamma)\mathbf{v}^r - \eta \frac{1}{s} \sum_{i \in S_r} \nabla f_i(\mathbf{x}_{i,r+1})$

---

# F   Additional Details on Experiments

## F.1   Convex Quadratics

We generate random vectors $\{b_{i,j}\}$ and diagonal matrices $\{A_{i,j}\}$ in the same way as in [22] such that $\max_{i,j}\{\|A_{i,j}\|\} = 100$ and $\delta \approx 5$. We use $n = 10$, $m = 5$ and $d = 1000$. We compare S-DANE and ACC-S-DANE with DANE. We use the standard gradient descent (GD) with constant stepsize $\frac{1}{200} \leq \frac{1}{2L}$ for all three methods as the local solver, where $L$ is the smoothness constant of each $f_i$. We use $\lambda = 5$ for all three methods. We use the stopping criterion $\|\nabla F_{i,r}(\mathbf{x}_{i,r+1})\| \leq \frac{\lambda}{2}\|\mathbf{x}_{i,r+1} - \mathbf{v}^r\|$ for our methods ($\mathbf{v}^r$ becomes $\mathbf{y}^r$ for the accelerated method). We use $\|\nabla \tilde{F}_{i,r}(\mathbf{x}_{i,r+1})\| \leq \frac{\lambda}{r+1}\|\mathbf{x}_{i,r+1} - \mathbf{x}^r\|$ for DANE.

## F.2   Deep Learning Tasks

We simulate the experiment on one NVIDIA DGX A100. We split the training dataset into $n = 10$ parts according to the Dirichlet distribution with $\alpha = 0.5$. We use SGD with a batch size of 512 as a local solver for each device. For all the methods considered in Figure 3, we choose the best number of local steps among $\{10, 20, \ldots 80\}$ (for SCAFFNEW, this becomes the inverse of the probability) and the best learning rate among $\{0.02, 0.05, 0.1\}$. For this particular task, it is often observed that using control variate makes the training less efficient [38]. The possible issue comes from the fact that local smoothness is often much smaller than local dissimilarity for this task. We here remove the control variate term on line 6 in S-DANE which is defined as $\langle \mathbf{x}, \nabla f_i(\mathbf{v}^r) - \nabla f_{S_r}(\mathbf{v}^r) \rangle$. Moreover, if we write the explicit formula for $\mathbf{v}^{r+1}$ on line 8, it becomes $\mathbf{v}^{r+1} = \gamma \mathbf{x}^{r+1} + (1 - \gamma)\mathbf{v}^r - \eta \frac{1}{s} \sum_{i \in S_r} \nabla f_i(\mathbf{x}_{i,r+1})$ with $\gamma \in [0, 1]$ and $\eta > 0$. We set $\gamma = 0.99$ and $\eta$ to be the local learning rate in our experiment. The method can be found in Algorithm 5. Note that the only difference between it and FEDPROX is the choice of the prox-center. The best number of local steps for the algorithms without using control variates is 70 while for the others is 10 (otherwise, the training loss explodes).

## F.3   Implementation

To implement Algorithms 1 and 2 (Algorithm 5 with option 2 is the same as Algorithm 1) , each device has the freedom to employ any efficient optimization algorithm, depending on its computation power and the local data size. At each communication round $r$, these local algorithms are called to approximately solve the sub-problems defined by $\{F_{i,r}\}$, until the gradient norm satisfies a certain accuracy condition that is stated in the corresponding

theorems. The server only needs to perform basic vector operations. Note that $G_r$ defined in those algorithms has a unique solution so that $\mathbf{v}^{r+1}$ can be explicitly derived (in the same form as line 9 in Algorithm 5).

# G  Impact Statement

This paper presents work that aims to advance the field of distributed Machine Learning. There are many potential societal consequences of our work, none of which we feel must be specifically highlighted here

