# OpenReview forum: "Stabilized Proximal-Point Methods for Federated Optimization"
_NeurIPS.cc/2024/Conference — NeurIPS 2024 spotlight_

### Official Review · Reviewer_TYfb · 2024-06-12

**Soundness:** 3
**Presentation:** 3
**Contribution:** 2
**Rating:** 6
**Confidence:** 3

**Summary:**

The authors of the paper extend the DANE algorithm to the stabilized-DANE (S-DANE) algorithm based on intuition from stabilized proximal point method. They also enhance the proposed S-DANE method with Monterio-Svaiter acceleration. The algorithms proposed by the authors allow for partial participation and various local solvers to some extent. Convergence analysis and experiments are provided.

**Strengths:**

1. The presentation of the paper is clear, logical and easy to follow.
2. The paper proposes two novel algorithms S-DANE and its accelerated version and provides convergence analysis of them, experiments are also provided to validate the claim of the paper.
3. The proposed algorithms allow for partial participation and various local solvers in some sense.

**Weaknesses:**

1. The partial client participation case of S-DANE and accelerated S-DANE both relies on the bounded gradient dissimilarity assumption, which is violated in many cases, and is not accurate enough when describing the effect of data heterogeneity. This raises concerns of the reviewer whether or not the algorithm is practical in the case of partial client participation.

2. The paper wants to address the communication bottleneck in FL setting by reducing the total number of communication rounds the algorithm needed to reach a certain accuracy level. However, it is also important to take into account the number of bits transferred from the client to the server as they are often constrained by the bandwidth. In fact, the terms communication complexity/efficiency appeared in the paper are quite misleading.

3. As the authors have suggested, the analysis is carried out in the case where each local objective $f_i$ is $\mu$-strongly convex, which is a little restrictive.

4. There are some typos (e.g., Line 265).

**Questions:**

1. In Table 1, why do the authors compare the number of communication rounds rather than the total amount of communication for each algorithm? Does all of those algorithms have the same communication overhead in each communication round? If the reviewer is not mistaken, some algorithms (such as SVRP) in the table work in the single client setting, is it fair to compare them directly to S-DANE in the full client participation setting?

2. For S-DANE (and its accelerated version), how to determine the proper stepsize? In the convergence guarantee, $\gamma$ depends on the unknown average Hessian dissimilarity constant $\bar{\delta}_s$ and the bounded gradient diversity constant $\zeta$. Could the author illustrate on how the two constants are estimated?

**Limitations:**

No.

---

> ### Author Rebuttal · Authors · 2024-08-06
>
> We thank the reviewer for the great evaluation of our paper. Your every comment is important to us. We did our best to
> understand and reply to your constructive feedback as follows:
>
> > W1
>
> Thanks for the great question. Here are some justifications.
>
> 1. Interestingly, we do not use any smoothness-related assumptions to prove the communication complexity. Therefore, the function classes also include Lipschitz and non-smooth functions. In this case, we believe using Definition 4 is reasonable.
>
> 2. If we further assume that the function $f$ is $L$-smooth, then we agree that Definition 4 can be relaxed. For instance, one can consider $\frac{1}{n}\sum_{i=1}^n ||\nabla f_i (x) - \nabla f(x)||^2 \le \zeta^2 + \beta ||\nabla f(x)||^2 \le \zeta^2 + 2\beta L (f(x) - f^\star)$ with $\beta > 0$, which captures the growth behaviour of the dissimilarity. The only change in the proof will appear in line 589 where we have an additional error term which can be canceled by choosing small $\gamma$ and large $\lambda$ that depend also on $\beta$ and $L$.
>
> 3. The interesting regime of considering partial client participation is when the number of clients $n$ is potentially very large. Then Definition 4 allows almost unbounded distribution of the gradient dissimilarity  (similar to the notion of 'stochastic gradient noise' for centralized SGD). Therefore, we believe using this assumption is still reasonable in practice. More detailed discussions can be found in Section 2 from [22] where the same assumption is used.
>
> > W2 \& Q1
>
> Thanks for this nice question! We refer to the attached pdf for the clarification of your concerns. Please let us know if there is anything unclear. We will add the discussions to our main manuscript.
>
> > W3
>
> Yes! However, note that our analysis works for standard convex functions ($\mu = 0$) as well. In the strongly convex case, when the function is of the form $f(x) = \frac{1}{n} \sum_{i = 1}^n \phi_i(x) + \frac{\mu}{2} || x ||^2$, we can rewrite it as $f(x) = \frac{1}{n} \sum_{i=1}^n f_i(x)$ with $\mu$-strongly convex $f_i(x) = \phi_i(x) + \frac{\mu}{2} || x ||^2$.
>
> In practice, the strong convexity often comes from adding explicit regularization of the local convex loss function (e.g. regularized logistic regression). Then each function typically is strongly-convex and $\mu$ is often known.
>
> We can actually propose the version of our method for composite functions $F(x) = f(x) + \psi(x)$ for which such an artificial split would be unnecessary but we do not do it for simplicity.
>
> > W4: typos
>
> Thanks for finding the typos!
>
> > Q2
>
> Thanks for asking this important question. We refer to the section of 'adaptive S-DANE with line-search' in the attached pdf for answering this question.
>
> We thank you again for your great review! If you agree that we managed to address all issues, please consider raising your score–this would totally boost our chance. If you believe this is not the case, please let us know and we will try our best to answer your every question!

---

> > ### Comment · Reviewer_TYfb · 2024-08-10
> >
> > Thanks for the response. The rebuttal has addressed most of my concerns, therefore I decide to raise my score.

---

> > > ### Author Response · Authors · 2024-08-12
> > >
> > > Many thanks for the support!

---

### Official Review · Reviewer_vDCp · 2024-06-12

**Soundness:** 3
**Presentation:** 3
**Contribution:** 3
**Rating:** 7
**Confidence:** 4

**Summary:**

An algorithm for distributed convex optimization with partial participation is proposed, under a similarity assumption.

**Strengths:**

The proposed algorithm has complexity Acc-S-DANE has claimed communication complexity O(sqrt(delta/mu)log(1/epsilon), for the first time. (I did not check the details of the proof but the main lines look correct to me).

**Weaknesses:**

* I want you to discuss your statement "Suppose we use the standard gradient descent as a local solver, then the number of gradient steps required to solve the subproblem increases across the iterations k". I don't think this is correct. In fact, you should discuss the 5GCS algorithm in Grudzien et al. "Can 5th Generation Local Training Methods Support Client Sampling? Yes!" AISTATS 2023. 5GCS is essentially a Point-SAGA algorithm with inexact computation of the proximity operators, and different strategies to solve these subproblems are discussed. The number of GD steps does not increase, you just pay a log(L/mu) factor for that in the complexity.

* You need to provide 2 tables: Table 1 in the full participation case, and a second table on existing algorithms for partial participation of $s$ clients. In particular, 5GCS and TAMUNA in "TAMUNA: Doubly Accelerated Federated Learning with Local Training, Compression, and Partial Participation" arXiv:2302.09832, 2023 have communication complexity $O\left(\sqrt{\frac{nL}{s\mu}}+\frac{n}{s}\right)\log(1/\epsilon)$.

* The complexity of Catalyzed SVRP reported in Table 1 is not correct. This value is for a different communication model, which counts "exchanging a vector between the server and one of the clients as a single communication step". So, if $s$ clients communicate, the complexity is multiplied by $s$, and $s=1$ gives the best complexity in this sense, since asynchronicity is encouraged. This is clearly different from measuring the number of communication rounds, in which $s<n$ *worsens* the complexity.

* You should discuss the paper Beznosikov et al. "Similarity, Compression and Local Steps: Three Pillars of Efficient Communications for Distributed Variational Inequalities", arXiv:2302.07615

* You should compare in your experiments to some of the methods mentioned above, in the two heterogeneous (delta = L) and homogeneous (delta<<L) regimes: does your method improve on them by exploiting the similarity, whereas other methods do not? (it might be that an algorithm also benefits from similarity, but has just not been studied under the similarity assumption in the literature).

* Definition 4 is very restricting, this means that the $f_i$ are all the same up to a linear difference. This is an auxiliary result but it is not very useful in practice.



typo in several places : mu-convex -> mu-strongly convex

The paper [4] has been published in ICLR.

**Questions:**

Do you assume that $\delta\geq \mu$? Because a complexity of $\sqrt{\delta/\mu}$ is a strong statement, it means that the algorithm converges in 1 round if $\delta \rightarrow 0$. FedRed, for instance, has $(\delta+\mu)/\mu$. Please check.

---

> ### Author Rebuttal · Authors · 2024-08-06
>
> We thank the reviewer for the great evaluation of our paper. Your every comment is important to us. We did our best to
> understand and reply to your constructive feedback as follows:
>
> > W1
>
> Thanks for the nice question.
>
> 1. This statement only (appears and) refers to the accuracy condition written in line 124. To satisfy this condition, if using GD, then the number of steps should depend on $k$, which is correct.
>
> 2. This condition is only stated for the original proximal point method (eq 2). It is well-known that the local computation has an unnecessary logarithmic dependency on either k or the final accuracy for this method, which can be improved by using more advanced variants [1,2,3].
>
> 3. 5GCS is different from the original proximal point method when $n=1$. In Table 1, many previous works (with proximal-point steps) also do not require increasing the number of GD steps, such as AccSVRS. These facts do not contradict this statement because these are different methods.
>
> 4. Acc-S-DANE is strictly better than 5GCS at least in the full-client participation settings.  **1)** the communication complexity depends on $\delta_A$ instead of $L$ for 5GCS  **2)** Even if $\delta_A = L$, the number of local steps is 1 instead of $\tilde{O}( \sqrt{L/\mu} )$ for 5GCS. 3) we prove convergence for the function value which is stronger than the squared distance for 5GCS. 4) it works well when $\mu \to 0$ while 5GCS seems not well-defined when $\mu = 0$.
>
> Nevertheless, 5GCS is an excellent method in some settings and we will add more discussion on it.
>
> [1] Svaiter B.F. and Solodov, M.V. A hybrid projection-proximal point algorithm. Journal of Convex Analysis. 1999.
>
> [2] Carmon, Yair, et al. Recapp: Crafting a more efficient catalyst for convex optimization. ICML 2022.
>
> [3] Ivanova, Anastasiya, et al. Adaptive Catalyst for Smooth Convex Optimization. International Conference on Optimization and Applications. 2021.
>
> > W2
>
> Thanks for the suggestion! However, it might be difficult to compare these two methods directly with our methods and others, as different settings are considered: for instance, 1) we consider cross-device setting where $n$ might be very large and each client is stateless (no control variates are stored), and hence SAGA-type methods are not applicable 2) certain first-order dissimilarity condition has to be assumed in this setting 3) as a result, our rates do not have a dependency on $\frac{n}{s}$ but instead on $\zeta^2$.
>
> We will add the rates and discussions of these two excellent methods in our manuscript.
>
> > W3
>
> Thanks for the great question. The complexity is correct. In Table 1, we report the number of communication rounds for all the methods including SVRP (for which the number of rounds is in the same order as the total number of vectors transmitted). We refer to the attached PDF where the settings and how SVRP works are explained in detail. The main metric of SVRP/SVRS is different and we separated these two algorithms in Table 1 and added Remark 6.
>
> > W4
>
> Thanks for the reference. This is a strong work. However, 1) The main target is the same as SVRP/SVRS, i.e. minimizing the total number of bits transmitted across rounds, which is different from this work. 2) It does not consider acceleration (while AccSVRS does) but uses compression to reduce the bits (while AccSVRS does not) and 3) in the end, it requires $\tilde{O}( n + \delta \sqrt{n} / \mu )$ communication rounds scaling with $n$.
>
> > W5
>
> Thanks for the interesting suggestion. The paper is based on the existing theoretical results. Studying if certain previous algorithms can exploit similarity is a bit irrelevant to the focus of this paper, which could instead be an interesting future work.
> Also in many cases, by tuning hyperparameters, some methods can outperform their theoretical upper bounds. For instance, Scaffnew and Scaffold have a similar performance by tuning stepsize while Scaffold has not been proven to achieve acceleration. But as you suggested, we will try to see if some of your mentioned methods can benefit from the similarity in our toy example.
>
> > W6
>
> 1. Interestingly, we do not use any smoothness-related assumptions to prove the communication complexity.
>     Definition 4 says the difference function is Lipschitz, and the original functions can be non-smooth.
>
> 2. Lipschitz functions are not just linear functions. The log-sum-exp is not linear but Lipschitz.
>
> 3. If we further assume that the function $f$ is $L$-smooth, then we agree that Definition 4 can be relaxed. For instance, one can consider $\frac{1}{n}\sum_{i=1}^n ||\nabla f_i (x) - \nabla f(x)||^2 \le \zeta^2 + \beta ||\nabla f(x)||^2 \le \zeta^2 + 2\beta L (f(x) - f^\star)$ with $\beta > 0$, which captures the growth behaviour of the dissimilarity. The only change in the proof will appear in line 589 where we have an additional error term which can be canceled by choosing small $\gamma$ and large $\lambda$ that depend also on $\beta$ and $L$.
>
> 4. The interesting regime of considering partial client participation is when the number of clients $n$ is potentially very large. Then Definition 4 allows an almost unbounded distribution of the gradient dissimilarity  (similar to the notion of 'stochastic gradient noise' for centralized SGD). Therefore, we believe using this assumption is still reasonable in practice. More detailed discussions can be found in Section 2 from [22] where the same assumption is used.
>
> > Typos and reference
>
> Thanks for finding the typos and we will update this reference.
>
> > Q1
>
> Many thanks. Yes, in Table 1, we reported the interesting regime $\mu \le \Theta(\delta)$ and we will add it to the text.
>
> We thank you for your great review and appreciate your help in improving the paper! If you agree that we managed to address your concerns, please consider raising your score–this would totally boost our chance. If you believe this is not the case, please let us know and we will try our best to answer your every question!

---

> > ### Comment · Reviewer_vDCp · 2024-08-08
> >
> > Thank you for replying to the points raised. Assuming you will make the appropriate changes in the paper, I think it can now be accepted, so I am raising my score to 7.

---

> > > ### Author Response · Authors · 2024-08-08
> > >
> > > Thank you very much for your trust and your support.

---

### Official Review · Reviewer_iL8h · 2024-07-12

**Soundness:** 3
**Presentation:** 3
**Contribution:** 3
**Rating:** 7
**Confidence:** 3

**Summary:**

The paper introduces a stabilized version of DANE (S-DANE). It replaces the proximal point step with an extragradient-type step. With the well-designed subproblem criterion, the number of local gradient oracle queries improves over DANE in logarithmic terms. It further combines Monteiro-Svaiter acceleration with S-DANE, which leads to the best communication complexity. The paper also considers partial client participation.

**Strengths:**

a) S-DANE eliminates the logarithmic term present in DANE.

b) Accelerated S-DANE achieves a communication complexity of \(O(\sqrt{\delta_A/\mu \log(1/\epsilon)})\), which surpasses the complexities in the existing literature.

c) The clarity of the manuscript's writing is good.

**Weaknesses:**

See "Questions".

**Questions:**

1. Apart from its resemblance to the extragradient method when \(\mu = 0\), what is the underlying motivation for considering equation (3)?

2. Regarding line 129 on page 4, the statement "such computation overhead cannot be avoided" seems imprecise. In the strongly convex case presented in [35] and both the convex and strongly convex cases described by Lan et al., 2023, the number of gradient steps required to solve the subproblem does not necessarily increase with \(k\).

Lan, G., & Li, Y. (2023). A Novel Catalyst Scheme for Stochastic Minimax Optimization. arXiv preprint arXiv:2311.02814.

3. Some suggested references for extragradient:


Korpelevich, G. M. (1976). The extragradient method for finding saddle points and other problems. Matecon, 12, 747-756.

Nemirovski, A. (2004). Prox-method with rate of convergence O (1/t) for variational inequalities with Lipschitz continuous monotone operators and smooth convex-concave saddle point problems. SIAM Journal on Optimization, 15(1), 229-251.

---

> ### Author Rebuttal · Authors · 2024-08-06
>
> We thank the reviewer for the great evaluation and the support of our paper!
>
> > Q1
>
> Thanks for the interesting question. The geometric meaning of equation (3) can be found in [1]. Previously, we derived this equation directly from the proof. Since we want to have the one-step recurrence of the form:
> $$ a\bigl( f(x^{r+1}) - f(x^\star) \bigr) + \frac{1+\mu a}{2} ||v^{r+1} - x^\star||^2 \le \frac{1}{2} ||v^r - x^\star||^2 ,  $$
> then we can first use convexity at the beginning of the proof:
> $$ a f(x^\star) + \frac{1}{2} ||v^r - x^\star||^2 \ge a \bigl( f(x^{r+1}) + \langle \nabla f(x^{r+1}), x^\star - x^{r+1}\rangle + \frac{\mu}{2} ||x^{r+1} - x^\star||^2 \bigr) + \frac{1}{2} ||v^r - x^\star||^2 . $$
> Then it is natural to set $v^{r+1}$ to be the minimizer of the right expression in $x^\star$, which is $(a\mu + 1)$- strongly convex. After that, we can approximately get the main recurrence.
>
> [1] Svaiter B.F. and Solodov, M.V. A hybrid projection-proximal point algorithm. Journal of Convex Analysis. 1999.
>
> > Q2
>
> Yes, this statement is wrong for accelerated proximal-point methods. Indeed, apart from your references, we also found several adaptive catalyst frameworks [2,3] that successfully remove the logarithmic dependency on $k$. Many thanks for pointing this out and providing the important reference.
>
> [2] Carmon, Yair, et al. Recapp: Crafting a more efficient catalyst for convex optimization. ICML 2022.
>
> [3] Ivanova, Anastasiya, et al. Adaptive Catalyst for Smooth Convex Optimization. International Conference on Optimization and Applications. 2021.
>
> > Q3
>
> We appreciate the reviewer for providing these nice references and we will add them to our manuscript!
>
> We thank the reviewer again for your great review and appreciate your help in improving the paper!

---

> > ### Comment · Reviewer_iL8h · 2024-08-12
> >
> > In the experiment involving deep learning, the control variable is omitted. Is this to avoid the need for communication when solving the subproblem? Additionally, when option 2 is applied in Algorithm 3, what is the approximate communication complexity? I noted in a discussion with another reviewer that theoretical analysis might be available for this case.

---

> > > ### Author Response · Authors · 2024-08-12
> > >
> > > Thanks for the question.
> > >
> > > The main purpose of using option II for some experiments is not to avoid the need for extra communications to obtain the vector $\nabla f_{S^r}(v^r)$ (the clients still need to communicate with the server to get $v^r$ and exchange $x_{i,r+1}$ and $\nabla f_i (x_{i,r+1})$.) Somehow for different deep learning tasks, option I and option II (using or not using control variates) behave differently, For this particular experiment of using ResNet, algorithms without adding control variates often perform better, But for language tasks, using control variates is often better.
> > >
> > > The rate of Algorithm 3 with option 2 and $s=n$ is similar to Algorithm 1 with $s=1$. By picking $\lambda = \frac{\zeta^2}{\epsilon}$, we get a deterministic rate of:
> > >
> > > $$
> > > f(\bar{x}^R) - f^\star \le \frac{\mu}{2 [(\frac{\mu}{\lambda}+1)^R - 1]} ||x^0 - x^\star||^2 + \frac{\epsilon}{2} .
> > > $$

---

### Official Review · Reviewer_QbRz · 2024-07-15

**Soundness:** 4
**Presentation:** 4
**Contribution:** 3
**Rating:** 7
**Confidence:** 5

**Summary:**

This paper considers the problem of distributed optimization under second-order similarity under (strong) convexity and smoothness. The paper proposes a new algorithm, Stabilized DANE, which (a) matches the best-known communication complexity under Hessian similarity while (b) requiring that local computation problems are solved only up to an approximately constant accuracy (i.e. not an accuracy that depends polynomially on $1/\epsilon$ where $\epsilon$ is the desired solution accuracy). The authors also consider partial participation, where only a subset of the clients is available at any given time, and also provide an accelerated version of Stabilized DANE.

**Strengths:**

- The paper is written clearly and the new algorithms are well-motivated. The proofs are easy to follow.
- The stabilization technique is elegant and clearly obtains both theoretical improvements (mostly in local computation complexity, as far as I can tell) and in practice (as shown in Section 5).
- The algorithms developed build on DANE, which is well-known and already a strong algorithm.

**Weaknesses:**

1. "It is necessary to assume a certain level of dissimilarity" (lines 208-212), I'm not actually sure that's _necessary_. There are certainly upper bounds without this assumption. Why is it necessary? Relaxed assumptions like expected smoothness are often enough even in this setting.
2. The complexities given for Acc-SVRS/Catalyzed SVRP in Table 1 are not consistent with how the other rates are presented in this work-- the $n$ factors should not be there, since in Table 1 for every other work full participation is assumed anyway (i.e. they should all have an $n$ factor multiplied). Which brings me to another point: What is the advantage of S-DANE over SVRS/SVRP? Is it only the higher efficiency of local computation?
3. The paper right now has a few different settings (full participation, partial participation) that contrast to other settings in prior work. Remark 6 attempts to explain the difference but I think this should be explained more clearly and placed in the main body of the paper.

**Questions:**

- Please address my concerns in the weaknesses section.
- Is the hessian similarity actually smaller than the smoothness over the training trajectory in the CIFAR10 optimization task?
- Can you derive theory for the variant of your method with no control variates that you use in the deep learning experiment? Does it just reduce to stabilized prox then?

**Limitations:**

N/A.

---

> ### Author Rebuttal · Authors · 2024-08-06
>
> We thank the reviewer for the great evaluation of our paper! Your every comment is important to us. We did our best to
> understand and reply to your constructive feedback as follows:
>
> > W1
>
> Yes, this sentence is confusing! We will rewrite it.
> - This sentence (that was first written in [22]) particularly refers to the setting where $n$ might go to infinity and each client does not store any vectors on its device (what people call 'stateless'). Then this sentence says to prove convergence, we need to assume a certain level of dissimilarity, which is correct. This sentence does not necessarily refer to using Definition 4.
> - When $n$ is sufficiently small, then we can use SAGA-type methods (considered in Scaffold) that ask each device to store certain control variates. Then yes, no dissimilarity assumption is required. Meanwhile, we need to assume the $L$-smoothness of the function and the final complexity depends on $L$.
> - In this paper, we consider the same setting as in paper [22] (with potentially very large $n$). Then as you mentioned, the question is if using Definition 4 is too strong.  Interestingly, we do not use any smoothness-related assumptions to prove the communication complexity. Therefore, the function classes also include Lipschitz and non-smooth functions. In this case, we believe using Definition 4 is necessary.
> - However, as you mentioned, if we further assume that the function $f$ is $L$-smooth, then Definition 4 can exactly be relaxed. For instance, one can consider $\frac{1}{n}\sum_{i=1}^n ||\nabla f_i (x) - \nabla f(x)||^2 \le \zeta^2 + \beta ||\nabla f(x)||^2 \le \zeta^2 + 2\beta L (f(x) - f^\star)$ with $\beta > 0$, which captures the growth behaviour of the dissimilarity. The only change in the proof will appear in line 589 where we have an additional error term which can be canceled by choosing small $\gamma$ and large $\lambda$ that depend also on $\beta$ and $L$.
>
> We will make this sentence clear and add more discussions to the manuscript according to your suggestion!
>
> > W2 \& W3
>
> Thanks for raising these points! We refer to the attached PDF for clarification of your concerns. (In Table 1, we report the number of communication rounds for all the methods including SVRP for which the number of rounds is in the same order as the total number of vectors transmitted.) Please let us know if there is anything unclear. We will move these discussions to the main paper as you suggested!
>
> > Q2
>
> This is a very interesting question. From our experiments, we saw that the estimated dissimilarity quantity defined as in Figure 3 ranges from $10^{-4}$ to $10^{-3}$ in the CIFAR10 task (and we choose $\lambda = 10^{-3}$ and the local learning rate $10^{-1}$ for S-DANE). Unfortunately, we did not record the local estimated smoothness quantity for our experiments. Empirically studying the relation between Hessian similarity and smoothness in different deep-learning tasks with various NN structures is indeed a very interesting future direction.
>
> > Q3
>
> Yes! The proof is similar to the one for S-DANE with $1$ client participation. Let us compare Algorithm 3 with $s= n$ and Algorithm 1 with $s = 1$. The main difference is that the former considers deterministic averaging and the latter considers sampling (in the proof, we need to take expectation for the latter which is similar to the deterministic averaging for the former). Therefore, if we study Algorithm 3, we have to similarly assume a certain first-order dissimilarity condition. We are not fully sure what you mean by stabilized prox. But yes, it can be seen as stabilized FedProx.
>
> We thank you again for your great review and we appreciate your support! If you agree that we managed to address all issues, please consider slightly raising your score–this would totally boost our chance. If you believe this is not the case, please let us know and we will try our best to answer your every question!

---

> > ### Comment · Reviewer_QbRz · 2024-08-07
> >
> > I thank the authors for their response.
> >
> > 1. "In Table 1, we report the number of communication rounds for all the methods including SVRP for which the number of rounds is in the same order as the total number of vectors transmitted" I have read your note, and I am not sure I agree with the statement that the number of transmitted vectors / round is the most important variable, this very much depends on the network type. In any case, one way of still communicating this information is to indicate in an additional column, explicitly, how many model-size vectors are communicated per communication round. As far as I understand, for your method, this would be "n" or "s" while for SVRP/SVRS it would be 2, on average (n comm per epoch + n comms at the end of each epoch). I think the table, as it exists currently, does not provide the full picture to the reader about the trade-offs between those different algorithms.
> >
> > 2. If the paper is accepted, can you redo the experiments and add the plots showing the local dissimilarity over the training trajectory as compared to the smoothness?
> >
> > 3. I use the term "stabilized prox" to refer to the method given by eqn (3) in your work, with $F_k$ being chosen stochastically.

---

> > > ### Author Response · Authors · 2024-08-07
> > >
> > > > 1.
> > > Thanks for the great suggestion! We will add one more column to specify the number of vectors communicated per round to Table 1. For our method, it is $n$. For SVRP, it would be 1 with probability $1 - \frac{1}{n}$ and $n$ with probability $\frac{1}{n}$. Yes, this would provide a better picture for the reader about the trade-offs between these algorithms.
> > >
> > > >2.
> > > Yes, we will add this comparison.
> > >
> > > >3.
> > > Thanks for the clarification. The stochastic version of eqn (3) is the same as S-DANE with $s=1$, which is also equivalent to Algorithm 3 with $s=1$.
> > >
> > > Thanks for your quick response and your constant help in improving the paper.

---

> > > > ### Comment · Reviewer_QbRz · 2024-08-07
> > > >
> > > > Thanks for your response as well. In light of the promised changes, I have increased my score.

---

> > > > > ### Author Response · Authors · 2024-08-07
> > > > >
> > > > > We really appreciate your support!

---

### Official Review · Reviewer_peSJ · 2024-07-23

**Soundness:** 2
**Presentation:** 3
**Contribution:** 3
**Rating:** 7
**Confidence:** 4

**Summary:**

The paper introduces a new variant of the existing DANE algorithm for federated learning. The paper integrates the stabilized proximal point method into DANE to form S-DANE. Convergence analyses are provided showing that S-DANE has the same rate as DANE but with better dependency on the communication round. The accelerated variant of S-DANE is also proposed which achieves better rates than S-DANE. Numerical experiments are able to show the advantage of S-DANE and Accelerated S-DANE compared to existing works.

**Strengths:**

- The related work discussion of the paper is good. I am able to see how the paper is positioned among existing works.
- I have not seen stabilized proximal update applied to federated learning (FL) so the idea in the paper appears to be new for FL.
- S-DANE and accelerated S-DANE are able to achieve the best-known rate for their corresponding setting.
- The claims are supported by theoretical analyses.
- The convergence results are done under both full and partial client participations.
- Extensive experiments showing the advantage of S-DANE/Accelerated S-DANE.

**Weaknesses:**

- The design of the algorithm is a little bit unclear, see questions below.
- Somehow I only see the full 8 algorithm in the left-most plot in Figure 2, while in the remaining 3 there are only about 4 algorithms displayed in the plots, i.e. I cannot see Snaffnew, AccGradSliding, DANE-GD, GD.

**Questions:**

1. In both algorithm 1 and algorithm 2, aseach device $i \in S_r$ performs the update in parallel, I do not see how $\nabla f_{S_{r}} (v^r)$ or $\nabla f_{S_{r}} (y^r)$ is computed since it requires the evaluation of the gradient of $f_j$ for $j \in S_r, j\neq i$?

2. In Figure 4, Scaffnew actually performs worse than Scaffold, this is somewhat surprising given the experiments in Mishchenko et al. (2022). What is your explanation on this?

**Limitations:**

The paper adequately discusses the limitation.

---

> ### Author Rebuttal · Authors · 2024-08-06
>
> We thank the reviewer for the great evaluation of our paper. Your every comment is important to us. We did our best to
> understand and reply to your constructive feedback as follows:
>
> >  W1 \& Q1
>
> Thanks for the question. The description can be found from line 158 to 163. Specifically, during each communication round $r$, the server first sends $v^r$ to the sampled clients. Then each client $i \in S_r$ computes $\nabla f_i(v^r)$ and sends this vector back to the server. Afterwards, the server computes $\nabla f_{S_r}(v^r) = \frac{1}{s}\sum_{i \in S_r} \nabla f_i(v^r)$ and sends $\nabla f_{S_r} (v^r)$ back to these clients. The same procedure was written in Section 4 in the Scaffold paper [1].
>
> > W2
>
> Thanks for the great observation. The remaining 3 plots are about partial-client participation. Indeed, we do not plot these four algorithms as partial-client participation is not considered in their original papers. We will make it clear in the manuscript.
>
> > Q2
>
> Thanks for the great observation. Note that Scaffold has two choices of control variates (cv). The cvs between Scaffnew and Scaffold are different in our experiments. For Scaffold, we use the same cv as DANE (option I in [1]) while for Scaffnew the cv is similar to option II in [1].
>
> 1. In the Scaffnew paper, they use the same cv for both Scaffnew and Scaffold. This is why they perform similarly (with tuned parameters).
>
> 2. So far, no results have shown that the cv for Scaffnew can achieve communication reduction under second-order similarity. However, the other cv (option I) can exploit similarity [2]. This is perhaps why we can observe a better performance of Scaffold in certain experiments.
>
> 3. Figure 4 is about deep learning. We expect that the cv for Scaffnew {option II} can be slightly less stable than another cv (option I).
>
> [1] Sai Praneeth Karimireddy, Satyen Kale, Mehryar Mohri, Sashank Reddi, Sebastian Stich, and Ananda Theertha Suresh. Scaffold: Stochastic controlled averaging for federated learning. ICML 2020.
>
> [2] Xiaowen Jiang, Anton Rodomanov, and Sebastian U Stich. Federated optimization with doubly regularized drift correction. ICML 2024.
>
> We thank you again for your great review! If you agree that we managed to address all issues, please consider raising your score–this would totally boost our chance. If you believe this is not the case, please let us know and we will try our best to answer your every question!

---

> > ### Comment · Reviewer_peSJ · 2024-08-12
> > **Responses to authors**
> >
> > I thank the authors for the responses. I believe my concerns have been fully addressed. I have adjusted the score. If the  paper gets  accepted, I would like these responses to appear in the revised version as well.

---

> > > ### Author Response · Authors · 2024-08-12
> > >
> > > Yes! Many thanks for the support.

---

### Author Rebuttal · Authors · 2024-08-06

We thank all reviewers for their constructive evaluations of our manuscript and we appreciate all the help from the reviewers for improving the paper.

In this work, we aim to develop federated optimization algorithms that 1) minimize the number of required communication rounds to reach the desired accuracy and 2) achieve high efficiency in local computation. These two represent central metrics in federated contexts, given the high cost of establishing connections between the server and the clients.

Specifically, we:
- developed novel algorithms: S-DANE (basic version) and Acc-S-DANE (accelerated version), which achieve the best-known communication complexity (in terms of the number of rounds) and local computation efficiency among all existing basic and accelerated methods (in the full client participation setting). This is achieved by using a more stabilized prox-center in the proximal step.
- further provided auxiliary results about partial client participation and using arbitrary stochastic local solvers, making them attractive in practice.
- provided a simple analysis for both algorithms. We derive convergence estimates that are continuous in the strong convexity parameter $\mu$.

Three reviewers asked about the setting and the SVRP algorithm. We refer to the attached PDF for clarifications.

After submission, we realize the following improvements can be made:
- The proof can be much simplified. We can completely remove the sequence $\\{ v_{i,r+1} \\}$ and only keep $\\{ v^{r+1} \\}$.
- We found an important reference [1] which we will include in the related work section. Our proposed method (S-DANE) recovers [1] for the special case of $n=1$. The paper [1] also contains a geometric explanation of why the stabilized proximal-point method has better local efficiency which could be of interest to the readers of our paper.
- Both of our algorithms can be made fully adaptive by using line-search (in the full-client participation setting). The details can be found in the attached PDF. This is, as far as we know, the first result of adaptive hyper-parameter tuning in the setting of exploiting second-order dissimilarity.

We anticipate an interactive discussion with you, and we will be most happy to answer any remaining questions.

[1] Svaiter B.F. and Solodov, M.V. A hybrid projection-proximal point algorithm. Journal of Convex Analysis, 6 (1):59–70, 1999.

---

### Decision · Program_Chairs · 2024-09-25

**Decision:**

Accept (spotlight)

**Comment:**

Paper presents a new Federated Optimization algorithm, S-DANE, for convex minimization. First authors identify that employing an extra-gradient step to proximal point method relaxes the tolerance required for proximal point step. They call this technique stabilized proximal point method. This is exploited to reduce the amount of local computations done on clients to constant number of steps as opposed to logarithmic number of steps in DANE algorithm. Additionally, this method can support arbitrary local client solver and partial client participation. Further they employ Monteiro-Svaiter acceleration in the servers to obtain accelerated converge and reduced overall communication complexity. This achieves new state of the art results. Finally, they provide both convex and non-convex deep learning experiments.

All reviewers agreed this is a well written paper. Contributions are clearly highlighted and explained. Paper is novel and relevant for this conference. During rebuttal period authors also found 25 year old paper which proposes the same idea for n=1 case. Reviewers noted that in partial client participation current theorem requires strict bounded gradient dissimilarity condition. Additionally, current theory also assumes all client objectives are strong convex. There we also certain imprecise statements and overclaiming in the initial manuscript. During the rebuttal phase authors managed to address some of these issues and impress upon the reviewers the value of their work. I ask the reviewers to modify the paper and experiemnts as promised to the reviewers.